



# Wind-field estimation for lidar-assisted control: A comparison of proper orthogonal decomposition and interpolation techniques

Esperanza Soto Sagredo[1], Søren Juhl Andersen[1], Ásta Hannesdóttir[1], and Jennifer Marie Rinker[1]

[1]Department of Wind and Energy Systems, Technical University of Denmark, Denmark

**Correspondence:** Esperanza Soto Sagredo (espa@dtu.dk)

**Abstract.**

This study presents and evaluates three wind field reconstruction methods for real-time inflow characterization, with potential applications in lidar-assisted wind turbine control. The first method applies a least-squares fit of proper orthogonal decomposition (POD) modes to lidar measurements (POD-LSQ). The second uses inverse distance weighting (IDW) interpolation across the rotor plane. The third, POD-IDW, applies the POD-LSQ fit to the interpolated field. The methods are tested under semi-realistic conditions derived from large-eddy simulations (LES), using a hub-mounted lidar sensor implemented in HAWC2 on the DTU 10 MW reference turbine. Measurements are extracted under varying inflow conditions. A rotor-effective wind speed estimate, combined with the known vertical shear profile from LES, serves as the baseline for comparison. Reconstruction performance is quantified using a global mean absolute error, evaluated across combinations of scan count, POD mode number, and lidar beam angle. Optimal parameters are selected based on the minimum error. To assess physical accuracy, reconstructions are compared against true wind speeds, evaluating the effects of probe volume averaging, multi-distance measurement selection, cross-contamination, and other sources of error. For optimal inputs, POD-IDW achieves the highest accuracy, reducing error by 45.5% compared with the baseline estimation, at 5.4 times the computational cost. IDW performs similarly (44.9%) with optimal inputs, while POD-LSQ achieves a 39.4% reduction with minimal overhead (7%). Spectral analysis shows that volume averaging and scanning strategies introduce low-pass filtering that attenuates high-frequency turbulence, while preserving low-frequency content more accurately than the baseline. Reconstruction quality strongly depends on the number and spatial distribution of lidar measurements and the number of retained POD modes. Although demonstrated under idealized conditions, the methods show strong potential for real-time applications. Future work should integrate these reconstructions with flow-aware controllers to evaluate fatigue load reduction, particularly at tower level.

## 1 Introduction

The upscaling of wind turbines has led to increasingly large and flexible rotors. While larger rotors average out small-scale fluctuations, they also increase sensitivity to spatio-temporal wind variability, which impacts both power production and structural loading (Angelou and Sjöholm, 2022). To mitigate these effects, advanced control strategies are needed to enhance performance while minimizing fatigue and extreme loads (Dong et al., 2021; Angelou and Sjöholm, 2022; Russell et al., 2024).



Lidar-assisted control (LAC) has emerged as a promising approach to reduce fatigue loads (Bossanyi et al., 2012; Guo et al., 2023; Fu et al., 2023; Russell et al., 2024), extreme loads (Schlipf and Kühn, 2008), and the levelized cost of energy (Scholbrock et al., 2016; Simley et al., 2018). Conventional LAC systems use nacelle-mounted lidars, which measure upstream wind via Doppler sensing and enable feedforward control by anticipating turbulence (Bossanyi et al., 2012; Schlipf et al.,
2015; Simley et al., 2018). However, nacelle-mounted lidars performance is hindered by blade blockage, causing data loss and increased uncertainty in wind field estimation (Schlipf et al., 2018; Angelou and Sjöholm, 2022). Mounting the lidar on the hub or spinner (hereafter hub-lidar) mitigates blockage and improves scan availability.

Lidar technology also influences data quality. Continuous-wave lidars focus on a single range, while pulsed-wave lidars use discrete time intervals to measure multiple distances along the line-of-sight (LOS) (Letizia et al., 2023). Therefore, pulsed
lidars exhibit higher coherence with the rotor-effective wind speed (REWS), as they capture a broader spatial extent (Kumar et al., 2015).

Among LAC strategies, feedforward collective pitch control is the most established. It adjusts all blades simultaneously by using REWS estimated as the averaged wind velocities measured by the lidar and projected into the longitudinal direction (Held and Mann, 2019), improving rotor speed regulation and reducing loads (Dunne et al., 2011; Canet et al., 2021; Fu et al.,
2023). However, reliance on spatially averaged REWS becomes less valid as rotor size increases.

To address this, feedforward individual pitch control adjusts each blade independently in response to localized wind. Approaches include combining REWS with horizontal and vertical shear profiles (Schlipf et al., 2010; Dunne et al., 2012) or measuring blade-level wind speeds at fixed radial and azimuthal positions (Dunne et al., 2012; Russell et al., 2024). Despite their benefits, these methods rely on simplified inflow assumptions and do not resolve spatio-temporal structures, which become
increasingly inadequate for large rotor diameters. Thus, there is a critical need for real-time, high-fidelity wind field reconstruction algorithms capable of resolving the spatial and temporal structures of the incoming wind, enabling more advanced LAC strategies and improved load mitigation across both tower and blade components.

High-fidelity reconstruction methods are therefore needed to capture the full wind field dynamics. Spectral techniques (Dimitrov and Natarajan, 2016; Rinker, 2022; Fu et al., 2022; Guo et al., 2022), CFD-based optimization (Bauweraerts and Meyers,
2021), and Bayesian estimation (Bauweraerts and Meyers, 2020) exist but are computationally intensive and not suited for real-time applications. Physics-informed machine learning approaches show promise for fast inflow reconstruction (Zhang and Zhao, 2021a, b), but lack demonstrated scalability for utility-scale turbines.

Proper orthogonal decomposition (POD) offers a computationally efficient model reduction technique by decomposing velocity fields into spatial modes and time-dependent coefficients that capture the flow's temporal evolution. It has been used in
wind energy to study turbine wakes for individual flow cases (VerHulst and Meneveau, 2014; Newman et al., 2014; Andersen et al., 2017; De Cillis et al., 2020). While early POD applications lacked predictive generality (Meneveau, 2019), Andersen and Murcia Leon (2022) introduced a global POD basis by combining multiple flow cases, allowing the basis to span a broader parameter space and enabling consistent physical interpretation across different flow conditions. More recently, Céspedes Moreno et al. (2025) evaluated the performance of a global basis in reconstructing wake aerodynamics, showing that the reconstruction
error decreases and converges as more cases are included in the dataset.





Focusing now on the use of lidar measurements in combination with POD, recent studies have explored both wake characterization and inflow reconstruction. In the context of wakes, Hamilton et al. (2025) applied POD to horizontal scans from nacelle-mounted lidars to identify coherent turbulent structures experienced by a turbine operating in the wake of an upstream rotor. For inflow reconstruction, Sekar et al. (2022); Kidambi Sekar et al. (2022) combined SpinnerLidar measurements with POD to estimate the incoming turbulent wind field. However, their approach relies on a complex and non-commercial lidar system (Mikkelsen et al., 2013; Herges et al., 2017), and it requires prior knowledge of the inflow, limiting its predictive capability. To address these limitations, Soto Sagredo et al. (2024a) proposed a least-squares fit of POD (POD-LSQ) method using hub-lidar data to estimate modal amplitudes in real time without requiring prior flow information. While promising, this approach was developed using idealized Mann-generated turbulence (Mann, 1998), and its robustness under realistic inflow conditions remains to be demonstrated.

Interpolation offers another approach for inflow reconstruction. Techniques such as kriging, inverse distance weighting (IDW), and cokriging are widely used in meteorology to estimate wind from sparse data (Luo et al., 2007; Joyner et al., 2015; Friedland et al., 2016). Similar methods have been applied to lidar-based wind field reconstruction (Chu et al., 2021; Bao et al., 2022), though they struggle to resolve unsteady 3D flows structures due to limited coverage and assumptions. Particularly, IDW is a widely used interpolation technique in geosciences, environmental science, and spatial data analysis (Bokati et al., 2022), and was also used by Soto Sagredo et al. (2024b) to reconstruct rotor-plane wind fields using hub-lidar data. While promising for real-time use, that study focused on idealized conditions and a single wind speed.

This study addresses the need for robust, real-time wind field reconstruction under varying inflow conditions. We evaluate three techniques: POD-LSQ, IDW, and a hybrid POD-IDW approach, comparing them against a REWS-based baseline that includes the vertical shear profile. Using LES-generated inflow and a numerical six-beam pulsed hub-lidar, we assess reconstruction accuracy and sensitivity to different input parameters across methods and inflow conditions. In particular, POD-LSQ shows promise for LAC due to its computational efficiency and spatial fidelity.

The paper is structured as follows: Section 2 describes the methodology, including LES inflow, lidar setup, and reconstruction methods. Section 3 evaluates the performance of each method across varying inflow conditions, analyzing their sensitivity to input parameters and measurement uncertainty to identify the optimal parameter combinations for each reconstruction approach. Section 4 discusses implications and limitations, while section 5 concludes the paper and outlines future work.

## 2 Methodology

To evaluate the wind field estimation techniques, synthetic lidar data are generated using high-fidelity inflow conditions from LES and a numerical hub-lidar sensor implemented on the DTU 10 MW reference wind turbine model (Bak et al., 2013) in HAWC2 v13.1 (DTU Wind Energy, 2024), which includes a flexible tower and a five degree tilt. The methodology is first summarized in the following paragraphs before being described in detail in the subsequent subsections.

An overview of the wind field reconstruction concept using pulsed hub-lidar technology is shown in Fig. 1. The multi-distance projected LOS velocities captured by the hub-lidar are mapped onto a reconstruction plane, where an estimation





method reconstructs the spatio-temporal wind field. This high-resolution inflow can potentially support advanced control strate-
gies by enabling real-time adaptation to turbulent structures and improving load mitigation.

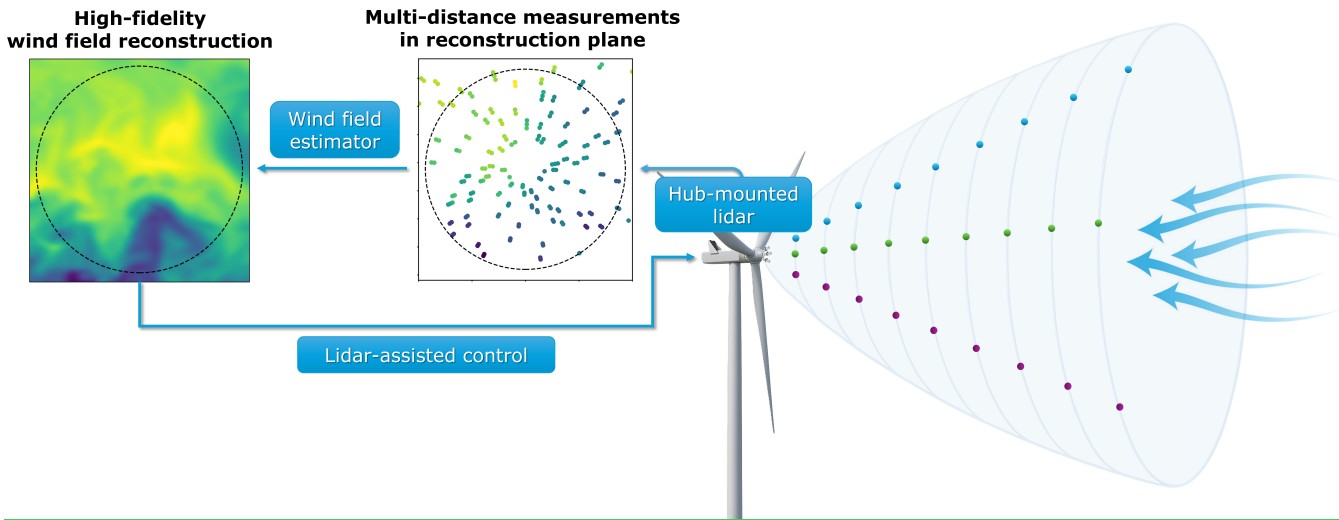

**Figure 1.** Schematic overview of the wind field reconstruction approach used in this study, using pulsed hub-lidar measurements for LAC.

Figure 2 outlines the numerical framework used to assess each method's accuracy. The primary goal is to quantify recon-
struction error and analyze sensitivity to key input parameters: the lidar beam half-cone angle ($\theta$), the number of scans ($n_{\mathrm{scan}}$),
the number of POD modes used for truncation ($K$), and the influence of measurement uncertainty.

The process begins with a turbulence database derived from LES inflow fields, divided into two subsets: Set A is used to
extract hub-lidar measurements in HAWC2, serving as the reference dataset, while set B is used to construct the global POD
basis. After preprocessing, the lidar data are mapped to the reconstruction plane and decomposed into wind fluctuations in the
YZ-plane, which are then used as inputs to the reconstruction methods. The reconstructed fields are compared against the refer-
ence fields to compute the reconstruction error, where the best-performing cases across multiple inflow conditions are identified
by selecting the parameter combinations that yield the lowest error. This systematic framework enables a consistent evaluation
of each method and supports the identification of robust configurations suitable for real-time wind field reconstruction.

## 2.1 LES precursor

The LES data used in this study originates from precursor simulations by Andersen and Murcia Leon (2022).

The precursor simulates a neutral atmospheric boundary layer driven by a steady pressure gradient over flat terrain. It
is performed using the EllipSys3D flow solver (Michelsen, 1992, 1994; Sørensen, 1995), which solves the Navier–Stokes
equations in general curvilinear coordinates using a finite volume method on a block-structured and collocated grid.

The computational domain spans 2880 m × 1440 m × 700.8 m, discretized into 576 × 288 × 320 cells, with uniform grid
spacing of 5 m in the longitudinal and lateral directions and 2.19 m vertically. To avoid spanwise locking of turbulent struc-





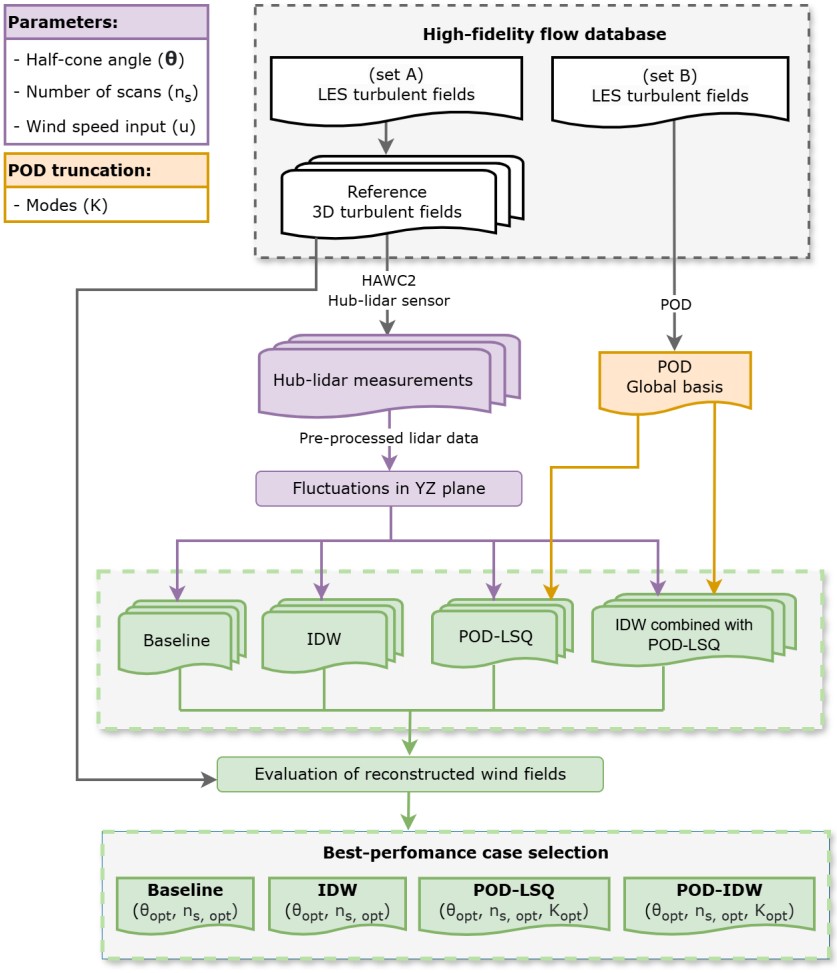

**Figure 2.** Numerical framework for wind field reconstruction evaluation using synthetic lidar measurements and four reconstruction techniques.

tures, a lateral shift is applied to the periodic boundaries in the longitudinal direction, following Munters et al. (2016). The simulation assumes a surface roughness of $z_0^{\mathrm{org}} = 0.05$ m and a friction velocity of $u_*^{\mathrm{org}} = 0.4545$ m s$^{-1}$. It runs for 82,600 s

(approximately 22.94 h) to ensure statistical convergence before collecting 28,800 s of inflow data.

As described by Castro (2007), neutral atmospheric boundary flows can be rescaled to generate multiple inflow conditions. We apply this rescaling following Troldborg et al. (2022):

$$u^{\mathrm{new}} = u_*^{\mathrm{new}} \left( \frac{u^{\mathrm{org}}}{u_*^{\mathrm{org}}} + \frac{1}{\kappa} \ln \frac{z_0^{\mathrm{org}}}{z_0^{\mathrm{new}}} \right), \tag{1}$$

where $\kappa = 0.41$ is the von Kármán constant, and the superscript "new" refers to the inflow condition to be generated, where

$u^{\mathrm{new}} \in [8.0, 12.0, 15.0, 18.0]$ m s$^{-1}$, obtaining four inflow wind speeds. All inflows have an average TI of approximately 11%.





## 2.2 LES datasets

The LES datasets (set A and B) are taken from two different locations within the LES domain and have distinct spatial dimensions in the lateral and vertical directions. Set A consists of $28800 \times 1 \times 61 \times 138$ grid points ($t \times x \times y \times z$), while set B includes $28800 \times 1 \times 39 \times 91$ grid points. Both datasets cover a 4-hour period at a temporal resolution of $d_t = 0.5$ s, with
spatial resolutions of 5 m in the lateral ($y$) direction and 2.19 m in the vertical ($z$) direction. An illustration of the locations in the LES domains is given in Fig. A in the appendix for clarity. For each inflow wind speed, the three turbulent velocity components ($u$, $v$, $w$) are extracted and rescaled following the method described in Sect. 2.1.

To ensure that the global POD basis characterizes the coherent turbulent structures over the rotor area, the spatial domain of set B is reduced and centered accordingly, spanning $190.3 \times 197.1$ m ($y \times z$). In contrast, set A retains a larger spatial extent
of $300.5 \times 300.0$ m ($y \times z$), providing sufficient coverage for aeroelastic simulation and lidar probe volume modeling.

From set A, sixteen non-overlapping 900 s segments are extracted to generate sixteen independent 3D turbulent inflow fields. These are transformed into spatial boxes under Taylor's frozen turbulence hypothesis (Taylor, 1938), which assumes that turbulent structures are convected downstream at a constant advection velocity $\bar{U}_o$—the average wind speed at hub height—without evolving in time. Under this transformation, the time and longitudinal directions are combined to define the longitudinal coor-
dinate $x$, with a spatial resolution given by:

$$d_x = \frac{\bar{U}_o T_{\text{sim}}}{N_x}, \quad \text{where } N_x = \frac{T_{\text{sim}}}{d_t} = 1800, \quad T_{\text{sim}} = 900 \text{ s.}$$

The resulting reference turbulence boxes have dimensions of $1800 \times 61 \times 138$ grid points ($x \times y \times z$), enabling accurate 15-minute HAWC2 simulations and ensuring full inclusion of lidar probe volume effects.

To account for HAWC2's transient effects, lidar initialization and boundary effects, the first 250 s and the last 50 s of the
simulation are discarded—beyond the conventional 100 s warm-up typically used in HAWC2—yielding the final reconstructed wind fields of 600 s, with dimensions of $1200 \times 39 \times 91$ grid points ($t \times y \times z$), with a time step $d_t = 0.5$ s.

## 2.3 Proper orthogonal decomposition and global basis

POD decomposes turbulent flow into orthogonal spatial modes that optimally capture the variance of the fluctuations (Lumley, 1967; Berkooz et al., 1993). Typically, POD is applied to a single flow case, yielding an orthogonal basis optimized for that
specific dataset.

To enable generalization across different flow conditions, Andersen and Murcia Leon (2022) utilized a "global" POD basis, which is derived by combining multiple cases, providing a more general representation across a broader parameter space. As the global POD modes are derived from multiple flow conditions, they are not optimized for any single case but instead capture generalized flow structures across the parameter space. However, as shown by Céspedes Moreno et al. (2025) the global basis
is still very effective and the suboptimality of a global basis compared to a "local" basis is at least an order of magnitude smaller than the truncation error.





To compute the basis, we first calculate the fluctuating component of the longitudinal velocity field by subtracting the temporal mean, $\bar{U}(y,z)$, from the full velocity field: $\boldsymbol{U}'(y,z,t) = \boldsymbol{U}(y,z,t) - \bar{U}(y,z)$. These fluctuations are then reshaped into column vectors over $N_t$ time steps for each of $N_c$ flow cases, forming the matrix $\mathbf{M} = [\boldsymbol{U}'_{1,1}, \ldots, \boldsymbol{U}'_{1,N_t}, \ldots, \boldsymbol{U}'_{N_c,1}, \ldots, \boldsymbol{U}'_{N_c,N_t}]$,

which is used to compute the POD modes using the randomized singular value decomposition (SVD) following the method of Halko et al. (2011).

The decomposition yields a set of orthonormal spatial modes $\mathbf{G} = [\boldsymbol{g}_1, \ldots, \boldsymbol{g}_{N_t-1}]$. A visualization of the first ten global POD modes is provided in Appendix B.

The corresponding modal time series are obtained by projecting the fluctuating flow onto the spatial POD modes using an

inner product $\phi_i(t) = \langle \boldsymbol{U}'(t), \boldsymbol{g}_i \rangle$.

A reduced-order approximation of the flow field can then be constructed as:

$$\boldsymbol{U}(y,z,t) = \sum_{i=1}^{K} \boldsymbol{g}_i(y,z)\,\phi_i(t) + \bar{U}(y,z) \tag{2}$$

where $K < N_t - 1$ is the number of retained modes and $\phi_i(t)$ are the modal time coefficients. This approach provides a low-dimensional representation of the flow, retaining dominant coherent structures while reducing computational complexity.

Although accurate representation of the flow physics in POD requires all three velocity components $(u,v,w)$ (Iqbal and Thomas, 2007), in this study we only extract the global POD modes using the longitudinal component $(u)$ from the LES dataset set B (see Sect. 2.2). This choice reflects the focus on reconstructing the longitudinal wind speed, which is the primary quantity of interest for LAC applications. The simplification is motivated by both methodological and practical considerations. First, $u$-component fluctuations dominate turbine loads (Dimitrov et al., 2018), whereas lateral $(v)$ and vertical $(w)$ components have

negligible impact (Dimitrov and Natarajan, 2016). Additionally, only the longitudinal $u$-component can be estimated from the LOS, due to the inability from the lidar system to measure 3D velocity but rather its projection onto the along-beam direction, known as the "cyclops dilemma" (Raach et al., 2014).

To avoid data leakage and promote generalization, the POD basis is computed from LES domains that are spatially offset from the reconstruction region (Appendix A).

## 175  2.4  Numerical lidar sensor

A numerical model of a hub-mounted pulsed lidar sensor is available in HAWC2 v13.1 (Soto Sagredo et al., 2023) to simulate realistic LOS wind measurements based on user-defined parameters for a single-beam lidar. This section outlines the sensor used in this study, including lidar parameters, coordinate system, and estimation of longitudinal wind speed.

The hub-lidar consists of a pulsed single-beam sensor installed on the spinner and constrained to the wind turbine model,

meaning that the lidar beams moves with the turbine structure and is affected by tower motion, including yaw, pitch, roll and structural vibrations. As the spinner rotates, the beam sweeps the rotor area, measuring LOS wind speeds by projecting the local $u$-, $v$-, and $w$-components onto the LOS direction (Fig. 1). Tower motion influences the measurements, introducing the relative velocity into the measured wind speed. Motion-induced fluctuations can be mitigated using frequency-domain filtering,





which preserves the integrity of the estimated wind field without introducing bias (Gräfe et al., 2023). Nevertheless, no motion

correction has been applied in the present analysis.

    Probe volume effects are simulated by HAWC2 using the weighting function and system parameters described in Meyer Forsting et al. (2017) for pulsed lidar systems. HAWC2 provides as outputs: (i) the probe volume-averaged LOS velocity $V_{\text{LOS, wgh}}$; (ii) the nominal LOS velocity $V_{\text{LOS, nom}}$, without volume averaging; and (iii) the corresponding measurement locations in $X, Y, Z$.

    The lidar beam unit directional vector, $\mathbf{n}(\theta, \psi, \omega(t))$, can be defined in a left-handed Cartesian coordinate system, with $X$
downwind and $Z$ vertical direction as:

$$\mathbf{n}(\theta, \psi, \omega(t)) = (-\cos(\theta), \; -\sin(\theta)\sin(\psi + \omega(t)), \; \sin(\theta)\cos(\psi + \omega(t))), \tag{3}$$

where $\theta$ is the half-cone angle, $\psi$ the azimuthal offset from blade 1 where the lidar beam is located (clockwise viewed downwind) and $\omega(t)$ is the azimuthal angle of blade 1 (origin aligned with the Z-axis upward, rotation clockwise viewed downwind).

    The LOS-velocity (also called radial velocity) can be mathematically expressed as:

$$V_{\text{LOS}}(\theta, \psi, \omega(t), \boldsymbol{P}) = \boldsymbol{n}(\theta, \psi, \omega(t)) \cdot \boldsymbol{U}(\boldsymbol{P}), \tag{4}$$

where $\mathbf{U}(\mathbf{P}) = [u(\mathbf{P}), v(\mathbf{P}), w(\mathbf{P})]^T$ is the wind vector, and $\mathbf{P}$ represents the location in space $(x, y, z)$ where the measurement is taken, as a function of the instantaneous hub location, the lidar beam unit vector and the range length, $f_d$.

    Figure 3 illustrates the beam configuration from three perspectives, for a beam with $\theta = 20°$, $\psi = 30°$, and a range length $f_d = 200$ m. The measurement point $\mathbf{P}$ is located at a distance $\overline{OP}$ from the lidar origin $\mathbf{O}$, placed at $(-L_s, 0, Z_{\text{hub}})$, where $L_s$
is the shaft length and $Z_{\text{hub}}$ the hub height. The spatial location of $\mathbf{P}$ can be expressed as:

$$\mathbf{P}(\theta, \psi, \omega(t), \beta, \nu, \gamma, f_d) = \mathbf{R}_{\text{hub}}(\beta, \nu, \gamma) \cdot \begin{bmatrix} -L_s \\ 0 \\ 0 \end{bmatrix} + \begin{bmatrix} 0 \\ 0 \\ Z_{hub} \end{bmatrix} + \mathbf{R}_{\text{hub}}(\beta, \nu, \gamma) \cdot f_d \cdot \mathbf{n}(\theta, \psi, \omega(t)), \tag{5}$$

where $\mathbf{R}_{\text{hub}}(\beta, \nu, \gamma)$ is the rotation matrix accounting for tilt ($\beta$), roll ($\nu$), and yaw ($\gamma$) of the wind turbine model for a left-handed coordinate system. For this study, $\beta = 5.0°$ and $\nu = \gamma = 0$.

    Therefore, the LOS-velocity accounting for beam orientation and tilt can be finally express as:

$$V_{\text{LOS}}(\theta, \psi, \omega(t), \beta, \mathbf{P}) = [\cos(\beta)\cos(\theta) - \sin(\beta)\sin(\theta)\cos(\psi + \omega(t))] \cdot u(\mathbf{P}) + [\sin(\theta)\sin(\psi + \omega(t))] \cdot v(\mathbf{P})$$

$$+ [\sin(\beta)\cos(\theta) + \cos(\beta)\sin(\theta)\cos(\psi + \omega(t))] \cdot w(\mathbf{P}) \tag{6}$$

The longitudinal wind speed is estimated by projecting $V_{\text{LOS}}$ onto the $x$-axis. This projection assumes negligible lateral and vertical components ($v$ and $w$) when the rotor is perfectly aligned with the wind (Schlipf et al., 2013; Simley et al., 2011), which introduces cross-contamination errors (Kelberlau and Mann, 2020). The final equation to project the LOS velocity to the longitudinal direction is thus as follows:

$$u_{\text{lidar}}(\theta, \psi, \omega(t), \beta, \mathbf{P}) = \frac{V_{\text{LOS}}(\theta, \psi, \omega(t), \mathbf{P})}{\cos(\beta)\cos(\theta) - \sin(\beta)\sin(\theta)\cos(\psi + \omega(t))}, \tag{7}$$

Effects related to optics, internal signal processing, and lidar-specific smearing are not considered in this study.





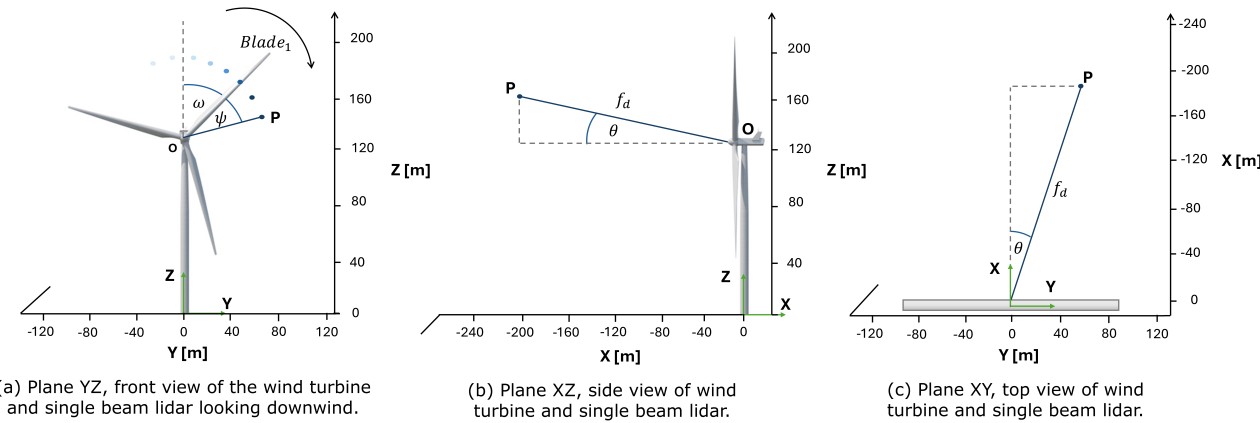

(a) Plane YZ, front view of the wind turbine and single beam lidar looking downwind.

(b) Plane XZ, side view of wind turbine and single beam lidar.

(c) Plane XY, top view of wind turbine and single beam lidar.

**Figure 3.** Left-handed coordinate system for a single-beam lidar mounted on the DTU 10 MW wind turbine model. Example with range length $f_d = 200$ m, half-cone angle $\theta = 20°$, azimuthal angle $\psi = 30°$, and blade 1 at $\omega = 35°$.

## 2.5 Lidar scanning strategy

The HAWC2 hub-lidar sensor supports flexible scanning configurations, allowing multiple beams with user-defined half-cone angles ($\theta$), azimuthal angles ($\psi$), and range lengths ($f_d$). In this study, a six-beam configuration is employed, with each beam

sampled sequentially at 5 Hz and no switching delay. The lidar records 29 fixed range gates spaced every 10 m, covering distances from 75 m to 355 m—consistent with common pulsed lidar practice (Peña et al., 2013). During post-processing, the beam sequence, sampling frequency, and switching delay can be customized.

Preliminary analysis (not shown for brevity) revealed that reconstruction accuracy improves when all six lidar beams are angled away from the central axis, rather than having a central beam pointing directly upwind. Although each beam samples

multiple range gates, a beam aligned with the wind direction collects data along a nearly straight line, with only a slight vertical shift due to turbine tilt. As a result, many measurements overlap and map into the same grid location, providing only a small number of central points in the fixed spatial grid. In contrast, angled beams increase their spatial coverage and provide more data for the estimation across the rotor plane. Similar results have been reported by Simley et al. (2014), where optimal scan radii was found to be approximately 70-75% of the rotor span.

The half-cone angle selection affects both projection errors—due to the assumption of negligible $v$ and $w$ components—and the spatial coverage of the rotor. Increasing the preview distance reduces cross-contamination and induction effects but increases errors due to wind evolution (Simley et al., 2012). Since HAWC2 does not model induction in the lidar sensor or temporal evolution of inflow, their impact is therefore not considered in this study.

The azimuthal angle defines each beam's angular separation from blade 1 in the rotating frame (Fig. 3), with allowable

values in this study, $\psi \in [0°, 120°, 240°]$, based on practical installation constraints. Specifically, for many multi-megawatt





wind turbines, manufacturers provide access to the hub through service hatches positioned every 120 degrees. Aligning the lidar beams with these access points simplifies both the installation and maintenance process.

The goal is to optimize the beam sequence to maximize rotor plane coverage over a given number of scans, where a "scan" is defined as the time needed to sample all six beams. At 5 Hz per beam, a single scan takes $t_{\text{scan}} = 1.2$ s.

Figure 4 illustrates the final six-beam hub-lidar configuration, which was found with an exhaustive search through iteration at fixed rotor speed $\Omega = 9.6$ rpm (rated speed of the DTU 10 MW turbine), yielding the optimal beam sequence: $\psi = [0°, 240°, 120°, 240°, 120°, 0°]$. Panel (a) shows the initial mounting azimuthal angles (YZ view), (b) the scanning trajectory over one scan (non-rotating frame), and (c) beam locations projected onto the XZ plane. This azimuthal configuration ensures optimal coverage of the rotor area above rated wind speed, since this is the operational range where LAC is used for load reduction.

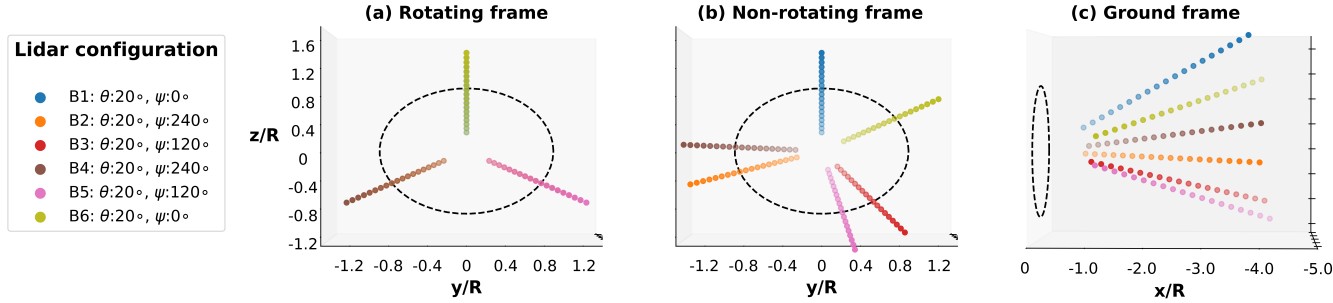

**Figure 4.** Six-beam lidar configuration with a half-cone angle $\theta = 20°$ for all beams, azimuthal angles $\psi = [0°, 240°, 120°, 240°, 120°, 0°]$, and 29 range-gates from 75 to 355 m spaced every 10 m. (a) Initial beam mounting positions (YZ view); (b) scanning trajectory over one scan at $\Omega = 9.6$ rpm; (c) beam locations projected on the XZ plane.

**2.6 Measurement selection for reconstruction**

Synthetic lidar measurements are extracted from the reference inflow fields (Sect. 2.2) using the hub-lidar sensor. During post-processing, the 20 Hz HAWC2 lidar output is downsampled to a 5 Hz sampling rate per beam, with no switching delay.

Measurements used for the reconstruction are selected through a spatial filtering process. In the lateral and vertical directions, the filter dimensions match those of the turbulence box from set A (refer to Sect. 2.2). In the longitudinal direction, the 245 filter spans a distance $d_{\text{span}} = \bar{U}_o \, n_{\text{scan}} \, t_{\text{scan}}$, where $n_{\text{scan}}$ is the selected number of scans and $t_{\text{scan}} = 1.2$ s is the duration of a single full scan. This longitudinal span is centered around the target reconstruction plane, located at $X_{\text{target}} = \bar{U}_o t_s$, where $t_s$ denotes the time step at which the reconstruction is evaluated. As a result, only measurements satisfying the condition $X_{\text{target}} - \frac{1}{2}\bar{U}_o \, n_{\text{scan}} \, t_{\text{scan}} < x < X_{\text{target}} + \frac{1}{2}\bar{U}_o \, n_{\text{scan}} \, t_{\text{scan}}$ are selected for the reconstruction. Additionally, we imposed a constraint requiring all selected measurements to be located at least one second upstream of the rotor plane, to ensure preview time.



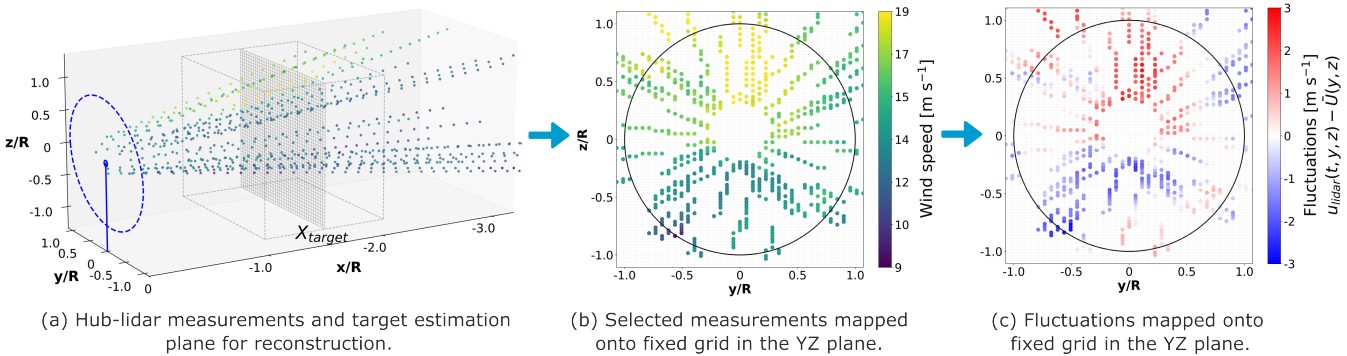

(a) Hub-lidar measurements and target estimation plane for reconstruction.

(b) Selected measurements mapped onto fixed grid in the YZ plane.

(c) Fluctuations mapped onto fixed grid in the YZ plane.

**Figure 5.** Lidar measurement selection for wind-field reconstruction at a single time step. (a) Lidar data advancing toward the rotor with the spatial filtering cuboid centered at $X_{\text{target}} = \bar{U}_o t_s$. (b) Mapping onto a fixed YZ grid. (c) Subtraction of the vertical wind speed profile.

This process is illustrated in Fig. 5, where (a) shows lidar data advancing towards the rotor with the cuboid centered around $X_{\text{target}}$, representing the spatial filtering region, and (b) maps selected measurements inside the cuboid onto the fixed YZ grid. If multiple measurements fall into the same grid cell, only the one closest to $X_{\text{target}}$ is retained. Panel (c) shows the resulting fluctuations after subtracting the known shear profile ($\bar{U}(y, z)$) from the LES dataset set A (refer to Sect. 2.2).

All data used in this study is gather in an publicly available dataset (Soto Sagredo et al., 2025a).

## 2.7 Wind-inflow reconstruction techniques

This section presents the methodologies used to reconstruct the longitudinal wind component ($u$) from synthetic lidar measurements obtained from the hub-lidar database (Sect. 2.6). Reconstructions are performed on a fixed $39 \times 91$ grid ($y \times z$), consistent with the global POD modes (Sect. 2.2). A total of 1200 time steps (600 s) are reconstructed with a sampling interval of $d_t = 0.5$ s.

All four methods use lidar-derived wind speed fluctuations as input, defined as $u'_{\text{lidar}}(t, y, z) = u_{\text{lidar}}(t, y, z) - \bar{U}(y, z)$, where $\bar{U}(y, z)$ is the known mean vertical wind speed profile (i.e., shear) from LES set A. This step standardizes the input across methods. After reconstructing the fluctuating component, the known shear is added back to obtain the full wind field for evaluation.

The four reconstruction techniques evaluated in this study are described below.

### 2.7.1 Baseline

The baseline methodology replicates the standard approach of estimating REWS from lidar measurements in LAC applications (Held and Mann, 2019). To ensure fair comparison, we account for the known shear profile. The reconstructed longitudinal wind speed at each time step is defined as:

$$u(t, y, z) = \frac{1}{n_{\text{meas}}} \sum_{i=1}^{n_{\text{meas}}} u'_{\text{lidar, i}} + \bar{U}(y, z) \tag{8}$$





where $u'_{\text{lidar},i}$ are the LOS-projected wind speed fluctuations (Sect. 2.4) within the circular region $A_R$ defined by $\sqrt{y_j^2 + (z_j - z_{\text{hub}})^2} \leq$ 95 m, centered at hub height. This area spans the rotor disk of the DTU 10 MW turbine with a 6 m margin to account for tower and shaft motion. The term $n_{\text{meas}}$ denotes the number of lidar measurements available within $A_R$.

### 2.7.2 Least-squares fit of POD modes

The Moore–Penrose pseudo-inverse is used to solve a least-squares problem for estimating the modal time series from lidar
measurements, originally introduced by Moore (1920) and independently by Penrose (1955).

Using the projected LOS fluctuations, $u'_{lidar}$, this field is stored in a matrix $\mathbf{M}_{\text{lidar}} \in \mathbb{R}^{N_y \times N_z}$ on a regular $YZ$ grid (Fig. 5c), with missing data points assigned NaN for numerical computational purposes, where $N_y = 39$ and $N_z = 91$ are the grid points across the lateral and vertical directions respectively. We define the index set $\mathcal{I} = \{i_1, i_2, \ldots, i_{n_{\text{meas}}}\}$ for grid locations with valid measurements, where $n_{\text{meas}}$ is the number of available measurements. Hence, the measurement vector $\mathbf{D}_{\text{lidar}}$ is defined
as $\boldsymbol{D}_{\text{lidar}} = \text{vec}(M_{\text{lidar}})_{\mathcal{I}} \in \mathbb{R}^{n_{\text{lidar}}}$, where $\text{vec}(\cdot)$ denotes column-wise vectorization.

The global POD modes $\mathbf{G}$ are sub-sampled at $\mathcal{I}$ locations, and the first $K$ global POD modes are selected to form the matrix $\mathbf{A} = \mathbf{G}_{\mathcal{I},1:K} \in \mathbb{R}^{n_{\text{meas}} \times K}$.

The modal coefficients $\tilde{\phi}$ are computed by solving the least-squares problem via pseudo-inverse of $\mathbf{A}$, by projecting $\boldsymbol{D}_{\text{lidar}}$ onto the column space of $\mathbf{A}$, leading to $\tilde{\phi} = (\mathbf{A}^\top \mathbf{A})^{-1} \mathbf{A}^\top \boldsymbol{D}_{\text{lidar}}$. The reconstructed wind field at each time step is then
computed as:

$$u(t,y,z) = \sum_{i=1}^{K} g_i(y,z)\tilde{\phi}_i(t) + \bar{U}(y,z), \tag{9}$$

where $g_i(y,z)$ is the $i$-th POD mode and $\tilde{\phi}_i(t)$ the corresponding estimated modal amplitude. Finally, the known mean profile $\bar{U}(y,z)$ is added back to recover the full field.

### 2.7.3 Interpolation with IDW

IDW estimates the wind fluctuations at a target location as a weighted average of nearby measurements, where influence decreases with distance. If the target coincides with a measurement location, the interpolated value matches the measurement. Mathematically, the interpolated longitudinal velocity at position $\mathbf{x} = (t,y,z)$ is calculated as:

$$u(t,y,z) = \frac{\sum_{i=1}^{n_{meas}} w_i(\mathbf{x}) u'_i}{\sum_{i=1}^{n_{meas}} w_i(\mathbf{x})} + \bar{U}(y,z), \tag{10}$$

where $u'_i$ is the wind speed fluctuation at measurement location $\mathbf{x_i}$, and the weights $w_i(\mathbf{x})$ are defined as:

$$w_i(\mathbf{x}) = \frac{1}{d(\mathbf{x}, \mathbf{x_i})^p}, \tag{11}$$

with $d(\mathbf{x}, \mathbf{x_i})$ the Euclidean distance between the location of unknown $u'$ and lidar measurements $u'_i$, and $p$ is a positive exponent controlling the decay rate, where higher values of $p$ emphasize nearer points. An exponent of $p = 3$ was found to minimize reconstruction errors.





### 2.7.4 Hybrid methodology: IDW combined with POD-LSQ

The final method combining IDW and POD-LSQ is presented in this section, called POD-IDW. First, the wind field fluctuations across the $YZ$ plane are estimated at each time step using IDW technique, following Sect. 2.7.3. Using the resulting IDW-reconstructed fluctuations (without the added vertical shear profile), the modal amplitudes $\tilde{\phi}$ are then estimated by performing a least-squares fit onto the global POD modes (Sect. 2.7.2).

Therefore, the modal coefficients $\tilde{\phi}$ are computed as $\tilde{\phi} = (\mathbf{A}^\top \mathbf{A})^{-1} \mathbf{A}^\top \boldsymbol{D}_{\text{IDW}}$, where $\boldsymbol{D}_{\text{IDW}} \in \mathbb{R}^{N_y N_z}$ is now the vectorization of the IDW reconstructed plane.

This hybrid approach is proposed to address a challenge encountered when using POD-LSQ: In areas without measurements, localized overfitting occurs. By using the IDW interpolated field to estimate the modal amplitudes, this limitation is mitigated.

### 2.8 Metrics for optimal parameter selection

Several parameters influence the accuracy of wind field reconstruction. To quantify the performance of the methods described in Sect. 2.7, we use the mean absolute error (MAE) computed within an area around the rotor, denoted as $A_R$ (see Sect. 2.7.1). The MAE is computed over the 10-minute simulations period ($N_t = 1200$, $d_t = 0.5$ s) as:

$$MAE_{\text{rotor},i,m}(\theta, n_{\text{scan}}, K) = \frac{1}{N_t} \sum_{t=1}^{N_t} \left[ \frac{1}{N} \sum_{(y,z) \in A_R} |u_{\text{ref},i}(t,y,z) - u_{i,m}(t,y,z)| \right], \tag{12}$$

where $N$ is the number of spatial grid points within the rotor area $A_R$, and $u_{\text{ref},i}$ denotes the reference wind field for inflow case $i$. The index $i \in \{1, \ldots, N_{\text{cases}}\}$ spans the set of inflow cases, with $N_{\text{cases}} = 64$, representing 16 independent 10-minute realizations across four wind speeds $\bar{U}_o \in \{8.0, 12.0, 15.0, 20.0\}$ m s$^{-1}$. The subscript $m$ represents each reconstruction method described in Sect. 2.7, with $m \in \{\text{Baseline, POD-LSQ, IDW and POD-IDW}\}$.

To enable fair comparison across different wind speeds, all reconstruction errors are normalized by the corresponding inflow mean wind speed $\bar{U}_{o,i}$. The global performance metric for a given method $m$ is defined as:

$$MAE_{\text{global},m}(\theta, n_{\text{scan}}, K) = \frac{1}{N_{\text{cases}}} \sum_{i=1}^{N_{\text{cases}}} \left( \frac{MAE_{\text{rotor},i,m}(\theta, n_{\text{scan}}, K)}{\bar{U}_{o,i}} \right) \tag{13}$$

The global reconstruction performance depends on several key parameters. First, the half-cone opening angle $\theta$ directly influences the spatial distribution and availability of lidar measurements across the rotor, as well as potential cross-contamination effects. In this study, we evaluate $\theta$ values in the range $\theta \in \{10.0°, 12.5°, \ldots, 50.0°\}$, using the same angle for all six beams.

Second, the number of lidar measurements per time step, $n_{\text{meas}}$, depends on the selected number of lidar scans, $n_{\text{scan}} \in \{1, 2, \ldots, 8\}$ (see Sect. 2.6). Increasing $n_{\text{scan}}$ provides higher data availability, but it can degrade reconstruction due to higher spatial filtering.

For POD-based methods, the number of retained modes $K$ is another important parameter. A higher $K$ allows for finer-scale flow reconstruction but can increase sensitivity to measurement sparsity and lead to overfitting. We evaluate $K \in \{10, 20, 30, 40, 50, 75, 100, 125, 150, 200\}$.





The optimal parameter combination for each method, $(\theta_{\mathrm{opt},m}, n_{\mathrm{scan,opt},m}, K_{\mathrm{opt},m})$, is defined as the one that minimizes the global normalized reconstruction error:

$$(\theta_{\mathrm{opt},m}, n_{\mathrm{scan,opt},m}, K_{\mathrm{opt},m}) = \arg\min_{\theta, n_{\mathrm{scan}}, K} \{MAE_{\mathrm{global},m}(\theta, n_{\mathrm{scan}}, K)\}, \tag{14}$$

where the search spans all 1,360 combinations in the discrete parameter space.

## 3 Results

To evaluate the performance and characteristics of the proposed reconstruction methods, we begin by assessing the accuracy of the modal amplitude estimation for POD-LSQ using lidar measurements, as discussed in Sect. 3.1. We then examine how the number of scans and modes influences the spectral content of the reconstructed inflow fields in Sect. 3.2. The effect of the half-cone opening angle on reconstruction accuracy is analyzed in Sect. 3.3, while Sect. 3.4 investigates the influence of wind speed quantity. Finally, Sect. 3.5 synthesizes these findings and presents a discussion on the optimal parameter configuration for each method.

To assess the influence of wind speed quantity selection has in reconstruction accuracy, we compare the reconstruction results using three wind speed quantities, hereafter referred to as: (i) the volume-averaged lidar estimate, $u_{\mathrm{lidar,\ wgh}}$, representing LOS velocities projected into the longitudinal direction and averaged over the lidar probe volume; (ii) the nominal lidar estimate, $u_{\mathrm{lidar,\ nom}}$, obtained from the same projection procedure but without applying volume averaging; and (iii) the true wind speed, $u_{\mathrm{fw}}$, corresponding to the reference longitudinal wind velocity extracted from the reference LES inflow field at the same grid locations in the $X_{\mathrm{target}}$ plane where the lidar measurements are fixed (see Fig. 5c) at each time step.

While only $u_{\mathrm{lidar,\ wgh}}$ represents a physically realistic lidar input, the alternative wind speed definitions are used for diagnostic purposes. In particular, $u_{\mathrm{fw}}$ isolates the performance of the reconstruction methods from key sources of measurement uncertainty, including volume averaging, cross-contamination, tower motion, and multi-distance fixed-plane mapping error. The latter refers to the spatial inconsistency introduced when lidar measurements—collected at different longitudinal positions—are projected onto a fixed estimation plane ($X_{\mathrm{target}}$), ignoring their true spatial separation along the longitudinal direction. In contrast, the difference between $u_{\mathrm{lidar,\ nom}}$ and $u_{\mathrm{lidar,\ wgh}}$ isolates the effect of volume averaging alone. These three wind speed definitions are used consistently throughout the paper to evaluate reconstruction accuracy and quantify the impact of measurement-related uncertainties.

### 3.1 Modal amplitude estimation with POD-LSQ

In real-time applications, reconstructing wind fields at each time step using global POD modes requires estimating the corresponding modal amplitudes, $\tilde{\phi}(t)$. This estimation is carried out using the POD-LSQ approach described in Sect. 2.7.2. In this section, we evaluate how wind speed quantity selection affects the accuracy of these modal amplitude estimations.

Figure 6 presents the first five modal amplitudes for a representative 10-minute inflow case with a mean wind speed of $\bar{U}_o = 15.35 \mathrm{\ m\ s^{-1}}$. The reference modal amplitudes (solid black lines) are compared against those estimated by POD-LSQ


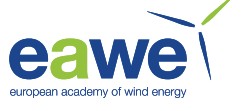


**Figure 6.** First five modal amplitudes estimated using POD-LSQ with $K = 50$ global POD modes, $n_{\text{scan}} = 7$ scans, and half-cone angle $\theta = 22.5°$. Top row: estimation using true wind speed ($u_{\text{fw}}$); second row: estimation using nominal lidar estimated ($u_{\text{lidar, nom}}$) wind speed; bottom row: estimation using volume-averaged lidar estimated ($u_{\text{lidar, wgh}}$) wind speed.

(dashed orange lines), using $K = 50$ global POD modes, $n_{\text{scan}} = 7$ scans, and a half-cone angle of $\theta = 22.5°$. The top row of Fig. 6 shows results obtained using the true wind speed as input, second row uses the nominal lidar estimate (without volume-averaged), while the bottom row uses the volume-averaged lidar estimate. This comparison allows us to isolate and quantify the impact of measurement errors on the estimation of modal amplitudes. Each subplot also reports the normalized MAE, defined as $\text{MAE}/\sigma$, where $\sigma$ is the standard deviation of the corresponding reference amplitude.

As expected, the estimation error $\text{MAE}/\sigma$ increases with mode number. Lower-order modes (e.g., Modes 1–3), which represent dominant large-scale structures, are reconstructed more accurately than higher-order modes (e.g., Modes 4–5), which correspond to finer-scale features. Furthermore, the use of the lidar-based estimates leads to consistently higher reconstruction errors compared to using the true wind speed. For mode 1, $\text{MAE}/\sigma$ increases by 55% when using nominal lidar estimate





compared to the true wind speed, while an increase of 41% is observed for volume-averaged lidar estimate. For mode 5, these
differences become more pronounced, with increases of 119% and 114%, respectively. These results highlight the sensitivity
of modal amplitude estimation to measurement-related uncertainties—such as cross-contamination, spatial offsets resulting
from the fixed-grid filtering approach used for multi-distance lidar measurements, and tower-induced motion. Higher-order
modes are particularly affected by these effects. Notably, probe volume-averaged helps to reduce such errors, as reflected in
the consistently lower $\mathrm{MAE}/\sigma$ values compared to the nominal case.

### 3.2 Sensitivity to number of scans and modes on the spectral content

To assess how the number of scans and modes affects the reconstruction of turbulent inflow fields, we analyze the power spectral
density (PSD) of the $u$-velocity fluctuations for a representative case with $\bar{U}_o = 15.35 \text{ m s}^{-1}$ over a 10-minute simulation. The
PSD are estimated using Welch's method with a Hamming window, six segments, and 50% overlap, applied over the area of
interest, $A_R$, across the rotor (refer to Sect. 2.7.1. A smooth PSD is calculated by averaging the PSDs at every lateral and
vertical point in the grid.

#### 3.2.1 Number of scans and estimation method

This section compares the PSD of the original flow to the estimated flow for different estimation methods and numbers of
scans. The number of modes used in the POD-based estimation methods is kept fixed for this analysis; the impact of POD
modes is analyzed in detail in Sect. 3.2.2.

Figure 7 presents the PSD results for (a) Baseline, (b) POD-LSQ, (c) IDW, and (d) POD-IDW, using $\theta = 22.5°$ and $K = 100$
global POD modes. Each panel shows reconstructions for various numbers of lidar scans ($n_{\mathrm{scan}} \in \{4, 8, 12, 16\}$) and two types
of wind speed inputs: the true wind speed ($u_{\mathrm{fw}}$, solid lines) and the volume-averaged lidar estimate ($u_{\mathrm{lidar, wgh}}$, dashed lines). For
reference, the full LES turbulence spectrum is plotted in solid black. The truncated POD spectrum (gray dashed) corresponds
to the projection of the LES flow onto the first 100 POD modes using the standard inner product, which is a better basis of
comparison for the POD methods. The vertical dashed line indicates the tower's first fore-aft eigenfrequency, $f_{\mathrm{tower}}$. The energy
drop-off beyond 0.1 Hz in the LES reference is attributed to grid resolution limitations, as reported in Thedin et al. (2023);
Doubrawa et al. (2019); Rivera-Arreba et al. (2022). Similarly, the further roll-off observed in the truncated POD spectrum is
due to modal truncation. As noted by Liverud Krathe et al. (2025), these high-frequency limitations do not significantly affect
fatigue analysis outcomes.

**Low frequencies**
At frequencies below 0.1 Hz, all reconstruction methods except the baseline (Fig. 7a) closely follow the LES spectrum when
estimating using the true wind speed $u_{\mathrm{fw}}$ (solid lines). IDW effectively reproduces the LES spectrum, while POD-based meth-
ods align with the truncated POD reference, consistent with their basis truncation. In contrast, reconstructions based on the
volume-averaged lidar estimate consistently exhibit lower energy from 0.02 Hz upwards. This reduced energy results from two
sources of spatial filtering: the intrinsic averaging within the probe volume (Peña et al., 2017) and the measurement selection
procedure, which maps multi-distance observations—taken at varying longitudinal positions—onto a fixed grid. This process



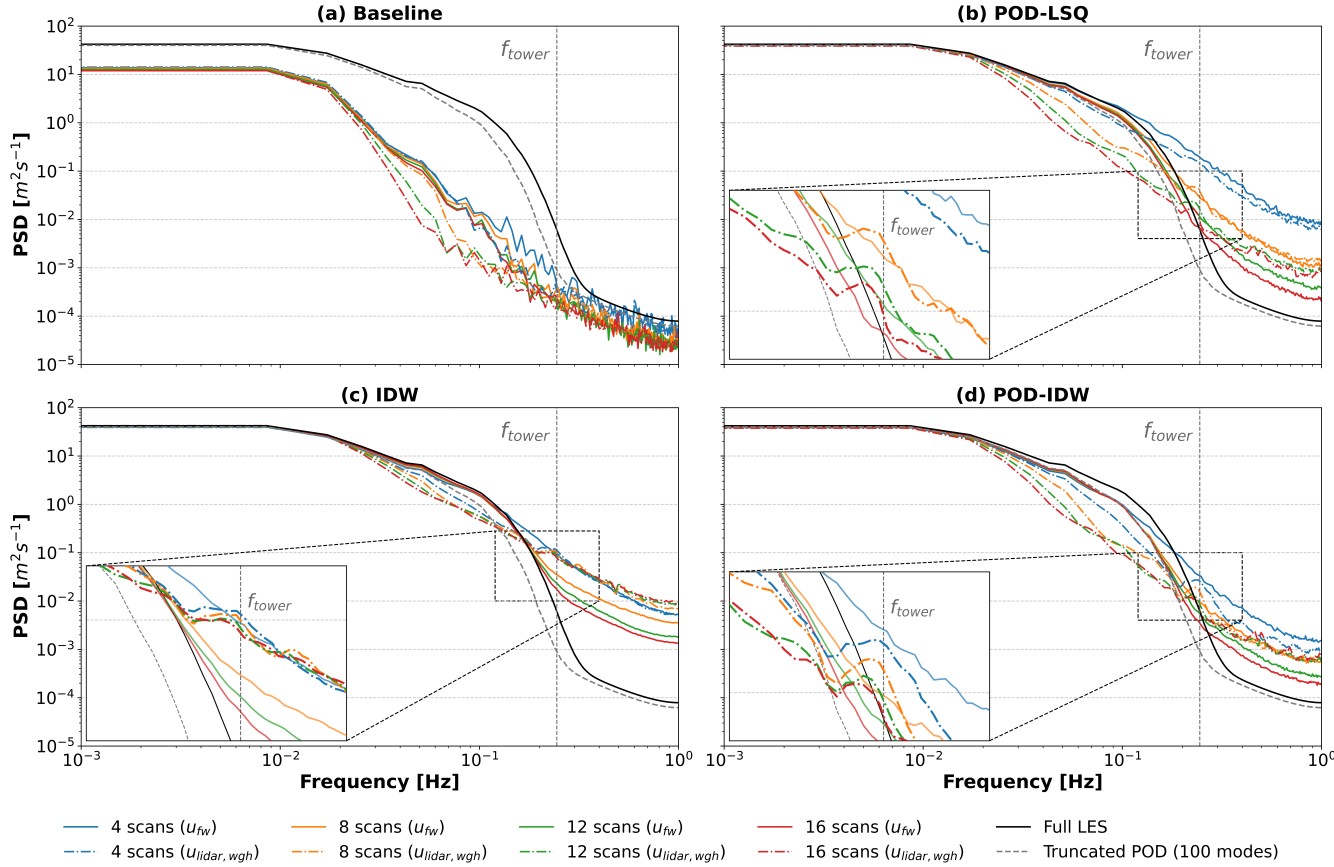

**Figure 7.** Influence of the number of scans $n_{scan}$ on the turbulence spectra of the $u$-component over a 10-minute simulation for each method, using $\theta = 22.5°$, $K = 100$ global POD modes for POD-based methods, for the true wind speed ($u_{fw}$) and volume-averaged lidar estimate ($u_{lidar,\,wgh}$) as input parameters for the reconstruction. Welch's method with Hamming window (six segments, 50% overlap) is applied and the PSDs around the area $A_R$ (see Sect. 2.7.1) around the rotor area averaged for smoothing.

smooths out turbulent fluctuations by blending information across different regions of the inflow. This filtering effect also impacts the baseline method, though to a lesser extent given its already simplified reconstruction approach.

**High frequencies**

In the high-frequency range, the flow estimated using POD-LSQ, POD-IDW, and IDW has more energy than the LES flow for both $u_{fw}$ and volume-averaged lidar estimate. When using $u_{fw}$, increasing the scan count reduces this extra spectral energy for these methods, drawing their spectra closer to the LES reference. This reduction in high-energy content with higher scans is driven by the number of available measurements in the rotor plane, which increases with $n_{scan}$. For example, in this study, four scans correspond to approximately 15% coverage of the $YZ$ plane, while 16 scans increases that coverage to around 42%.

Having more measurements in the rotor plane reduces overfitting for the POD methods, which results in lower spectral energy





for more scans. In IDW, a higher number of measurements increases spatial coverage, reducing gaps between data points. This leads to more accurate interpolation and lower reconstruction error.

However, for lidar-based inputs, the measurement selection procedure described in Sect.2.6 selects data points across varying longitudinal positions and maps them onto a fixed grid, which becomes increasingly extended as more scans are included. This

introduces an additional spatial filtering effect, resulting in a sharper energy drop beyond 0.017 Hz. While IDW does not exhibit the same steep decline, this instead reflects increased noise due to higher multi-distance fixed-plane mapping errors, rather than improved reconstruction fidelity. Unlike POD-based methods, which enforce spatial coherence through a global modal basis, IDW does not incorporate spatial correlations between measurements, which can compromise accuracy, especially for irregularly distributed data (Li et al., 2020; Bokati et al., 2022). Furthermore, IDW assumes isotropic flow variations and is

sensitive to outliers, making it more susceptible to errors caused by spatial separation. This behavior is further illustrated in the time series example for $n_{\mathrm{scan}} = 16$ shown in Appendix C.

**Tower natural frequency**

A secondary effect visible in Fig. 7 is the presence of a peak near the tower's natural frequency, $f_{\mathrm{tower}} = 0.25$ Hz, in the volume-averaged lidar estimate reconstructions (zoomed area). This is caused by tower-induced motion distorting the lidar

measurements. The effect is particularly visible when $n_{\mathrm{scan}} = 4k$ for $k \in \mathbb{N}$, since four scans approximately matches the tower's oscillation period. The peak is more pronounced in POD-based reconstructions and less distinguishable in IDW due to IDW's elevated background spectral energy near $f_{\mathrm{tower}}$.

### 3.2.2 Number of POD modes

This section demonstrates the impact of number of POD modes on the estimation for the POD-LSQ and POD-IDW methods,

where $n_{\mathrm{scan}}$ is kept constant.

Figure 8 presents the PSD for (a) POD-LSQ and (b) POD-IDW for different numbers of modes, $K \in \{50, 100, 150, 200\}$, using $n_{\mathrm{scan}} = 4$, $\theta = 22.5°$, and the volume-averaged lidar estimate as input. Lower values of $K$ capture the large-scale, low-frequency content of the flow, while increasing $K$ introduces more high-frequency energy. At low frequencies, POD-LSQ aligns more closely with the truncated POD spectrum, whereas POD-IDW shows greater deviation due to the influence of

the initial IDW plane, which introduces interpolation-related errors. At higher frequencies (above 0.1 Hz), POD-LSQ tends to overestimate energy, primarily due to overfitting in the modal amplitude estimation, as mentioned in Sect 3.2.1. This effect will be discussed again in Sect. 3.3. POD-IDW mitigates overfitting by using the IDW-reconstructed plane as input for estimating modal amplitudes, which smooths the high-frequency content and reduces overfitting.

### 3.2.3 Summary and methodological implications

The scan count, $n_{\mathrm{scan}}$, exerts a dual influence on reconstruction quality. On the one hand, increasing $n_{\mathrm{scan}}$ improves fidelity when using ideal inputs ($u_{\mathrm{fw}}$) by enhancing spatial coverage. On the other hand, it also increases the longitudinal separation between measurements, which can degrade reconstruction accuracy when lidar-based inputs ($u_{\mathrm{lidar, wgh}}$) are used due to amplified multi-distance fixed-plane mapping error.





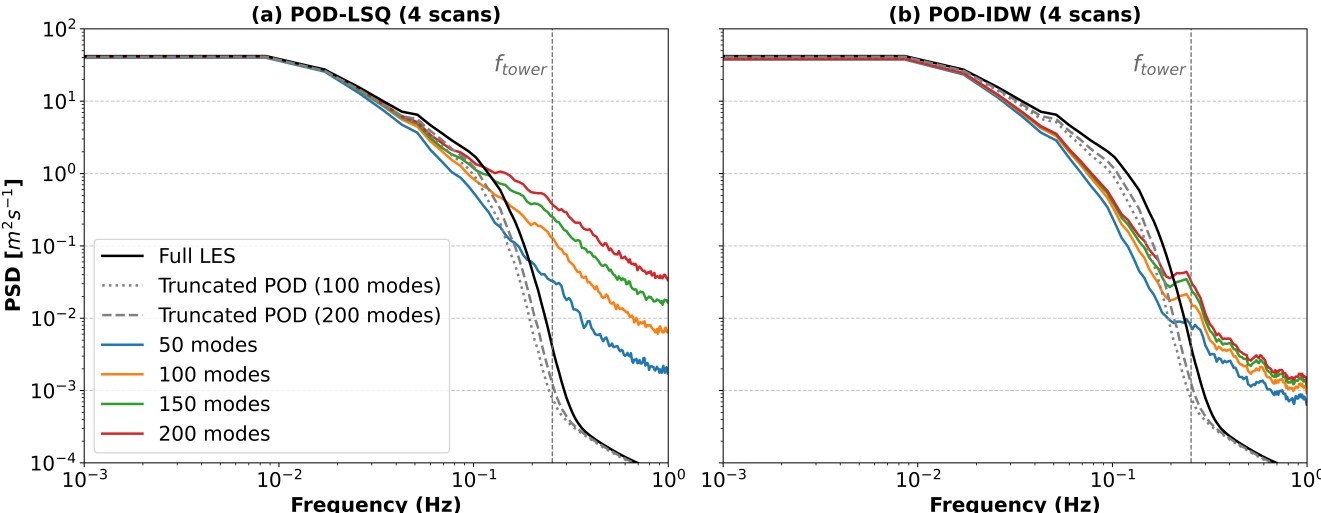

**Figure 8.** Influence of the number of POD modes ($K$) on the turbulence spectra of the $u$-component over a 10-minute simulation for POD-LSQ (a) and POD-IDW (b), with $K \in 50, 100, 150, 200$ and $n_{\text{scan}} = 4$. Welch's method is applied (six segments, 50% overlap) over the interest area $A_R$ (see Sect. 2.7.1).

Each reconstruction method responds differently to this trade-off. IDW is particularly sensitive to spatial separation, as it does not account for spatial correlation across measurements. POD-LSQ, by contrast, is more affected by the number and distribution of available measurements—especially at higher values of $K$—since accurate estimation of modal amplitudes requires adequate spatial coverage. POD-IDW, which combines an initial IDW interpolation with subsequent POD fitting, inherits some limitations from the interpolation step but benefits from the modal projection, which helps to smooth errors introduced by spatial filtering (refer to Appendix C). As a result, the optimal selection of both scan count ($n_{\text{scan}}$) and number of POD modes ($K$) should be tailored to the specific sensitivities of each method and the nature of the available input.

Finally, attention should be given to the effects of tower motion, which introduce spurious energy near the tower's natural frequency ($f_{\text{tower}}$). These artifacts, particularly prominent in lidar-based reconstructions, may require correction techniques or frequency-domain filtering to avoid negative impacts on both control performance and aeroelastic load assessments.

## 3.3 Effect of half-cone angle on reconstruction performance

The selection of the half-cone angle, $\theta$, affects reconstruction accuracy in two main ways: (1) by influencing the number and spatial distribution of lidar measurements across the rotor plane, and (2) by increasing cross-contamination in the volume-averaged lidar estimate derived from LOS measurements.

Figure 9 illustrates these effects at a representative time step across the $YZ$ rotor plane. Each column corresponds to a half-cone angle $\theta \in \{10.0°, 17.5°, 20.0°, 22.5°, 35.0°, 50.0°\}$ using $K = 200$ modes and $n_{\text{scan}} = 4$ for all reconstructions. Row (a) shows the location of the lidar measurements, and each point's color represents the difference between the true wind speed

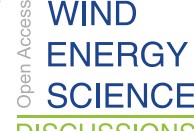

($u_{\text{fw}}$) and the volume-averaged lidar estimate ($u_{\text{lidar, wgh}}$), while rows (b)–(e) present reconstruction errors relative to the LES reference for (b) Baseline, (c) POD-LSQ, (d) IDW, and (e) POD-IDW. MAE values are reported above each case.

**Figure 9.** Difference across the $YZ$ plane between the reconstructed ($u(t,y,z)$) and reference ($u_{\text{ref}}(t,y,z)$) wind fields, for $\bar{U}_o = 15.35$ m s$^{-1}$ at a single time step. The columns correspond to increasing half-cone angles from $10°$ to $50°$, using four scans and 200 POD modes, where (a) shows the difference between the hub-lidar volume average projected LOS measurements ($u_{\text{lidar, wgh}}$) and the true wind speed ($u_{\text{fw}}$), and (b)–(e) present the differences between the reconstructed and LES reference cases for (b) baseline, (c) POD-LSQ, (d) IDW, and (e) POD-IDW. Blue colors indicate underestimation, while red colors indicate overestimation of wind speeds relative to the reference field.





As shown in Fig. 9a, at $\theta = 10.0°$, measurements concentrate near the upper center of the rotor due to turbine tilt, yielding $n_{\text{meas}} = 457$. With increasing $\theta$, spatial coverage improves as measurements spread more broadly across the rotor plane. How-
ever, beyond a certain point, outer beams extend beyond the rotor, reducing central coverage. For example, at $\theta = 50.0°$, only
182 valid measurements remain as outer range gates extend beyond the rotor area due to beam inclination. Additionally, larger
$\theta$ values increase cross-contamination—indicated by the more intense coloring of the points—due to increased misalignment
with the line-of-sight and the longitudinal turbulence.

Figure 9b–e shows how each method responds to changes in $\theta$. The baseline method is relatively insensitive, exhibiting
consistent performance across all angles. POD-LSQ, in contrast, is highly sensitive, as accuracy depends on both the num-
ber and spatial distribution of measurements. At $\theta = 10.0°$, performance degrades despite a high measurement count due to
poor spatial coverage and localized overfitting. At $\theta = 50.0°$, the number of measurements is too low ($n_{\text{meas}} = 182$) to sup-
port fitting $K = 200$ modes, yielding an underdetermined system and degraded accuracy. These results highlight that, for
POD-based methods, ensuring $n_{\text{meas}} \geq K$ is necessary but not sufficient—broad spatial coverage is equally critical. Notably,
although $\theta = 10.0°$ yields more measurements than $\theta = 35.0°$, POD-LSQ performs worse, underscoring the importance of
spatial distribution over raw measurement count.

The sensitivity of IDW and POD-IDW to $\theta$ is less significant than for POD-LSQ. However, both methods exhibit increased
reconstruction error at $\theta = 50.0°$, consistent with greater cross-contamination in the LOS-derived wind speed (Fig. 9a). POD-
IDW inherits interpolation errors from the IDW field, so spatial inaccuracies propagate into the final reconstruction. Still,
applying POD over the IDW field reduces spatial inconsistencies, resulting in smoother fields and improved performance over
IDW alone—though the improvement in MAE is limited (Fig. 9d-e).

Careful selection of the half-cone angle is therefore essential, particularly for POD-based methods. The angle must balance
spatial coverage with minimal contamination from $v$ and $w$ components in the LOS signal. This selection depends on lidar
geometry, turbine tilt, and alignment between beam orientation and the rotor plane. It is also critical to ensure that $n_{\text{meas}} \geq K$,
and that measurements are sufficiently distributed to avoid overfitting in POD-LSQ. The baseline method, by contrast, remains
largely insensitive to $\theta$, provided enough data are collected across the scan radius (Simley et al., 2018).

## 3.4 Impact of wind quantity on reconstruction performance

To assess how lidar-induced measurement errors affects reconstruction accuracy, we evaluate each method using the three
wind speed inputs: (i) true wind speed ($u_{\text{fw}}$), (ii) nominal lidar estimate without volume averaging ($u_{\text{lidar, nom}}$), and (iii) volume-
averaged lidar estimate ($u_{\text{lidar, wgh}}$).

Figure 10 shows the global reconstruction error, $MAE_{\text{global}}$ (defined in Sect. 2.8), as a function of the half-cone angle $\theta$ for
each method using its optimal configuration, $n_{\text{scan,opt}}$ and $K_{\text{opt}}$, determined as shown in Fig. 2 and listed in Table 1. Results are
presented for four methods—(a) Baseline, (b) POD-LSQ, (c) IDW, and (d) POD-IDW—under all three wind speed inputs. The
performance spread across these inputs reflects the influence of measurement errors and volume averaging.

Consistent with Sect. 3.3, the baseline method (Fig. 10a) shows almost identical results regardless of input type, due to its
rotor-averaging approach that inherently smooths spatial uncertainties. Despite this robustness, it consistently yields the highest





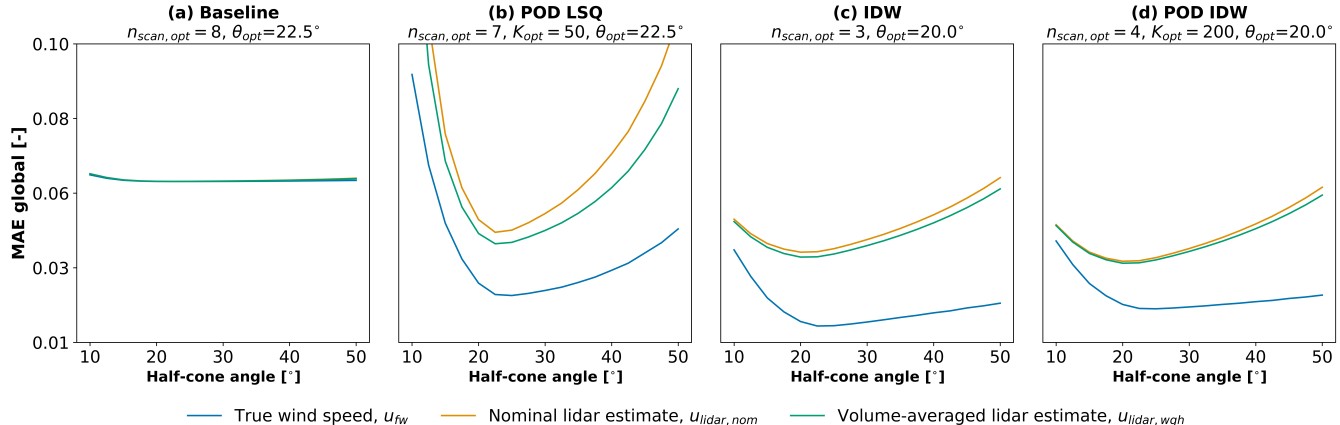

**Figure 10.** Global mean absolute error, $MAE_{\text{global}}$, as a function of half-cone angle, $\theta$, for the best-performing configurations: (a) baseline, (b) POD-LSQ, (c) IDW, and (d) POD-IDW. Results are shown using: true wind speed ($u_{\text{fw}}$, blue), nominal estimate ($u_{\text{lidar, nom}}$, orange), and volume-averaged estimate ($u_{\text{lidar, wgh}}$, green).

reconstruction error. Among the other methods, POD-LSQ remains the most sensitive to half-cone angle selection, with errors increasing sharply as $\theta$ deviates from the optimum due to overfitting and cross-contamination, which impair modal amplitude estimation (see also Sects. 3.2 and 3.3). In contrast, IDW and POD-IDW show less sensitivity to $\theta$, with errors remaining relatively stable even as measurement distribution deteriorates. For POD-IDW, this robustness is partly inherited from IDW, as it uses the interpolated IDW field for modal fitting and thus avoids overfitting (see Sect. 3.3).

IDW shows the largest performance gap between $u_{\text{fw}}$ and lidar-based inputs, highlighting its strong sensitivity to uncertainties—particularly the multi-distance fixed-plane mapping error and cross-contamination. POD-IDW performs slightly better, as the modal decomposition enforces spatial coherence, mitigating some interpolation-related errors. POD-LSQ also benefits from modal constraints, reducing sensitivity to input uncertainty, though it remains highly dependent on $\theta$.

Across all methods, reconstructions using the true wind speed $u_{\text{fw}}$ significantly outperform those from the two line-of-sight quantities, but the volume-averaged input ($u_{\text{lidar, wgh}}$) outperforms the nominal lidar estimates ($u_{\text{lidar, nom}}$). In other words, there is a significant increase in error caused by measurement inaccuracies, but adding volume averaging actually reduces the error. This demonstrates the benefit of modeling the probe volume, as volume averaging reduces cross-contamination and improves reconstruction quality—a trend especially evident for POD-LSQ and consistent with the findings reported in Soto Sagredo et al. (2025b).

In summary, reconstruction accuracy is significantly influenced by measurement quantity. IDW is the most affected, followed by POD-IDW, which inherits interpolation errors. POD-LSQ shows better resilience but depends strongly on appropriate angle selection. The baseline, though robust to measurement variations, performs worst overall. These results emphasize the importance of accounting for probe volume effects and tuning method-specific parameters for reliable lidar-based wind field reconstruction.





## 3.5 Overall method performance

The ultimate conclusion we would like to draw is which of the investigated methods performs "best." The optimal parameter sets—half-cone angle $\theta_{\text{opt}}$, number of scans $n_{\text{scan,opt}}$, and number of POD modes $K_{\text{opt}}$—are those that minimize the global mean

absolute error $MAE_{\text{global}}$ for each reconstruction method (Sect. 2.8). These values are summarized in Table 1 for both the true wind speed input ($u_{\text{fw}}$) and the volume-averaged lidar estimate ($u_{\text{lidar, wgh}}$), with corresponding performance trends shown in Fig. 10.

**Table 1.** Optimal parameter combinations ($\theta_{\text{opt}}, n_{\text{scan,opt}}, K_{\text{opt}}$) that minimize the global mean absolute error, $MAE_{\text{global}}$, for each reconstruction method, using as inputs the true wind speed ($u_{\text{fw}}$) and volume-averaged lidar estimated wind speed ($u_{\text{lidar, wgh}}$).

| Method | Wind speed input | $\theta_{\text{opt}}$ | $n_{\text{scan, opt}}$ | $\bar{n}_{\text{meas}}$ | $K_{\text{opt}}$ | $MAE_{\text{global}}$ | Error compared to baseline |
|---|---|---|---|---|---|---|---|
| **Baseline** | True wind | 22.5° | 8 | 868 | - | 0.0549 | -0.04 % |
| | Volume-averaged | 22.5° | 8 | 868 | - | 0.0549 | Reference case |
| **POD-LSQ** | True wind | 25.0° | 8 | 791 | 200 | 0.0206 | -58.8 % |
| | Volume-averaged | 22.5° | 7 | 776 | 50 | 0.0328 | -39.4 % |
| **IDW** | True wind | 22.5° | 8 | 868 | - | 0.0120 | -74.5 % |
| | Volume-averaged | 20.0° | 3 | 403 | - | 0.0288 | -44.9 % |
| **POD-IDW** | True wind | 25.0° | 8 | 791 | 200 | 0.0183 | -65.6 % |
| | Volume-averaged | 20.0° | 4 | 518 | 200 | 0.0287 | - 45.4 % |

Using lidar-based input, POD-IDW achieves the lowest reconstruction error, with a 45.4% reduction in $MAE_{\text{global}}$ compared to the baseline. IDW performs nearly as well (44.9% reduction), followed by POD-LSQ (39.4%). With true wind input, IDW

achieves the best accuracy (74.5% reduction), followed by POD-IDW (65.6%) and POD-LSQ (58.8%). For the baseline, only a negligible 0.04% difference exists between input types—reflecting its robustness to uncertainty but also its limited resolution, which results in the highest reconstruction error overall. Note that the baseline method achieves its best performance with $n_{\text{scan}} = 8$, although using fewer scans results in nearly identical accuracy. This insensitivity to scan count reflects the inherent spatial averaging of the baseline approach.

Measurement errors—including probe volume averaging, cross-contamination, multi-distance fixed-plane mapping error, and tower motion—degrades reconstruction accuracy for all methods except the baseline. This effect is most pronounced for IDW (Sects. 3.2–3.4), where the error reduction drops from 74.5% (with $u_{\text{fw}}$) to 44.9% (with $u_{\text{lidar, wgh}}$), and the optimal number of scans decreases from eight to three. These shifts reflect IDW's strong sensitivity to the assumption that all selected measurements lie on the same plane, neglecting their actual longitudinal location in space. For all methods, using true wind

input generally shifts the optimal half-cone angle $\theta$ upward by 2.5–5°, as cross-contamination is no longer a limiting factor. POD-LSQ, in particular, improves its performance by increasing the number of modes from 50 to 200 with only one additional scan, highlighting the impact of measurement uncertainty on modal amplitude estimation (Sect. 3.1).





Overall, POD-IDW offers the best reconstruction accuracy with lidar-based inputs but comes with a higher computational cost. IDW is simpler and moderately expensive, but does not capture spatial correlations and assumes isotropic flow vari-
ations, leading to unrealistic estimates under certain conditions. POD-LSQ provides a good balance between accuracy and efficiency but requires careful tuning of lidar configuration and POD parameters. The baseline method is the fastest and most robust to measurement uncertainty, yet consistently delivers the poorest reconstruction quality. Ultimately, reliable wind field reconstruction depends on accounting for measurement uncertainty and appropriately tuning method-specific parameters.

## 4  Discussion

The goal of this study was to assess the reconstruction accuracy and robustness of three methods under semi-realistic inflow conditions, using LES-generated data and a hub-mounted lidar simulator. Evaluation was based on a global metric, $MAE_{\mathrm{global}}$, which quantifies the deviation from the true inflow field across multiple conditions. While effective for identifying optimal parameters and quantifying deviations from the reference wind field, this metric does not capture the ability of the reconstructed fields to drive realistic turbine dynamics. A more robust analysis would involve evaluating the reconstruction error on turbine
load channels, which is the subject of future work.

The current study makes several simplifying assumptions due to limitations in the available tools. Notably, the HAWC2 lidar implementation does not account for turbulence evolution (Bossanyi, 2013; de Maré and Mann, 2016) or induction effects (Borraccino et al., 2017; Mann et al., 2018), as it is based on Taylor's frozen turbulence hypothesis and BEM-based inflow dynamics. As a result, the inflow is treated as a stationary free-stream field, from where the lidar measurements are
extracted. Although this is not realistic, it provides a controlled environment to evaluate reconstruction accuracy. Future work should incorporate 4D inflow fields from LES simulations that include these effects to better evaluate how they influence reconstruction performance—and how they may be compensated in practice.

All methods in this study also rely on knowledge of the mean shear profile to reconstruct the flow. This allows a consistent comparison across methods and avoids introducing additional shear estimation uncertainties. Although accurate shear esti-
mation from hub-lidar data is feasible over longer time frames, following similar procedure to the one described in Eq.(4) from Sebastiani et al. (2022), it was not the focus of this study. Importantly, POD-based methods require subtraction of the mean flow to eliminate trends during the estimation of the modal amplitudes. In contrast, IDW and the baseline approach can operate directly on lidar measurements without prior shear estimation. Furthermore, this study focused on neutral boundary layer conditions with a turbulence intensity around 11%, which are representative but do not encompass the full range of field
scenarios. Extending the evaluation to different atmospheric conditions—including stable and unstable stratification—would broaden validation.

The real-time performance of the proposed reconstruction methods was evaluated to confirm their suitability for online applications. On a standard PC using a Python implementation and input from four lidar scans ($\approx$520 measurements), the baseline method required $\approx$19 ms per time step. POD-LSQ introduced minimal overhead, increasing computational time by
only 7% (to $\approx$20 ms). In contrast, IDW was more computationally demanding, requiring 4.5$\times$ the baseline duration ($\approx$86 ms),




due to the cost of the interpolation step. POD-IDW, which combines interpolation with a subsequent POD fitting, was the most expensive at ≈103 ms per time step (5.4 × the baseline). Despite the higher cost for IDW and POD-IDW, all methods remained within practical real-time constraints.

The methods proposed in this study are not well suited for nacelle-mounted lidar systems—the most common configuration 575 for wind turbine-mounted lidars (Letizia et al., 2023)—due to their limited spatial resolution and blade blockage, which reduce the number of available measurements. The accuracy of methods like POD-LSQ depends on having a high number of spatially distributed inputs. Moreover, the hub-lidar scanning pattern in this study was optimized for the rated rotor speed of the DTU 10 MW turbine; other turbines or operating conditions would require re-optimization to ensure adequate coverage.

Finally, real-world lidar systems also face additional uncertainties, such as optical misalignment, Doppler noise, signal 580 processing errors, and probe volume smearing. It is also critical to ensure sufficient lead time for control while accounting for latency, memory constraints, and turbulence advection. Future work should address these challenges through adaptive filtering and selection strategies (Schlipf, 2016), validated in conjunction with flow-aware control.

## 5 Conclusions

This study proposed and evaluated three methodologies for real-time reconstruction of wind inflow fields across the full rotor 585 plane, using lidar measurements extracted from LES simulations via a numerical hub-mounted lidar model in HAWC2. The methods include: POD-LSQ, which fits lidar data to a global POD basis using least squares; IDW, which interpolates the flow using inverse distance weighting from lidar data; and POD-IDW, a hybrid that estimates modal amplitudes from the IDW-reconstructed field. All were benchmarked against a baseline rotor-averaged approach based on conventional REWS estimation, a standard practice for LAC applications, that accounts for the mean known shear across the rotor plane.

When optimally configured, all proposed methods significantly outperformed the baseline, offering improved spatial resolution and turbulence reconstruction. POD-IDW achieved the lowest reconstruction error, reducing $MAE_{\text{global}}$ by 45.4% compared to the baseline estimation, followed by IDW (44.9% reduction) and POD-LSQ (39.4% reduction). All methods met real-time computational requirements. On a standard PC with four lidar scans (≈520 measurements), the baseline required ≈19 ms per time step, while POD-LSQ added only 7% overhead. IDW and POD-IDW required ≈86 ms and ≈103 ms, respec- 595 tively, but remained within practical limits.

Reconstruction performance was found to depend strongly on the number and spatial distribution of measurements, half-cone angle, measurement uncertainty, and the number of POD modes. POD-LSQ was especially sensitive to half-cone angle due to the tradeoff between number of measurements in the rotor plane and measurement coverage, which can lead to overfitting with higher numbers of modes. For POD-based methods, increasing the number of global modes $K$ increased the ability to 600 capture flow energy and improve inflow reconstruction, but required $K \leq n_{\text{meas}}$ and good coverage around the rotor to ensure numerical stability. POD-IDW relaxed this constraint by using a full interpolated field, supporting higher $K$ values at the cost of propagating interpolation errors and increased computational demand.





Measurement uncertainty had a notable impact, particularly for IDW and POD-IDW, which reconstruct the full plane directly from lidar data. IDW was most affected by multi-distance fixed-plane mapping errors and performed best with fewer scans due to its lack of spatial correlation modeling. The marginal accuracy gain between POD-IDW and IDW (0.5%) reflects the trade-off between reduced overfitting and inherited interpolation error. POD-LSQ also exhibited sensitivity, as uncertainties propagated through the modal amplitude fitting process. Overall, POD-LSQ offers the best compromise between reconstruction accuracy, computational efficiency, and robustness to lidar-related uncertainties—provided that adequate spatial coverage and a sufficient number of measurements are available to support the selected number of POD modes. By projecting lidar measurements onto a set of spatial patterns derived from POD, the method captures the dominant flow structures. This enables reliable spatial reconstruction even under imperfect measurement conditions, making POD-LSQ particularly well suited for real-time wind field estimation in LAC applications.

Future work should investigate the effects of rotor induction and turbulence evolution on reconstruction accuracy, as these are not captured in the current setup. Additionally, evaluating the proposed methods under a broader range of atmospheric stability conditions and turbulence intensities will help further define their robustness and operational limits. To fully assess their control relevance, these reconstruction techniques should also be coupled with a flow-aware controller and tested within a feedforward individual pitch control framework, enabling quantification of potential load reductions and operational benefits.





## Appendix A:  LES wind speed profile

A representation of the LES wind speed profile of the precursor is presented in Fig. A1, illustrating the two data sets used in
this study, where set A represent the location from where the reference 3D turbulence fields used in HAWC2 for the hub-lidar
generation were extracted, while set B shows the section from where the data to computed the global POD basis was extracted.
Noticed that the lateral distance ($Y$) between the center of the two boxes is 751.3 m. These two datasets are referred in the
overview of the methodology presented in Fig. 2.

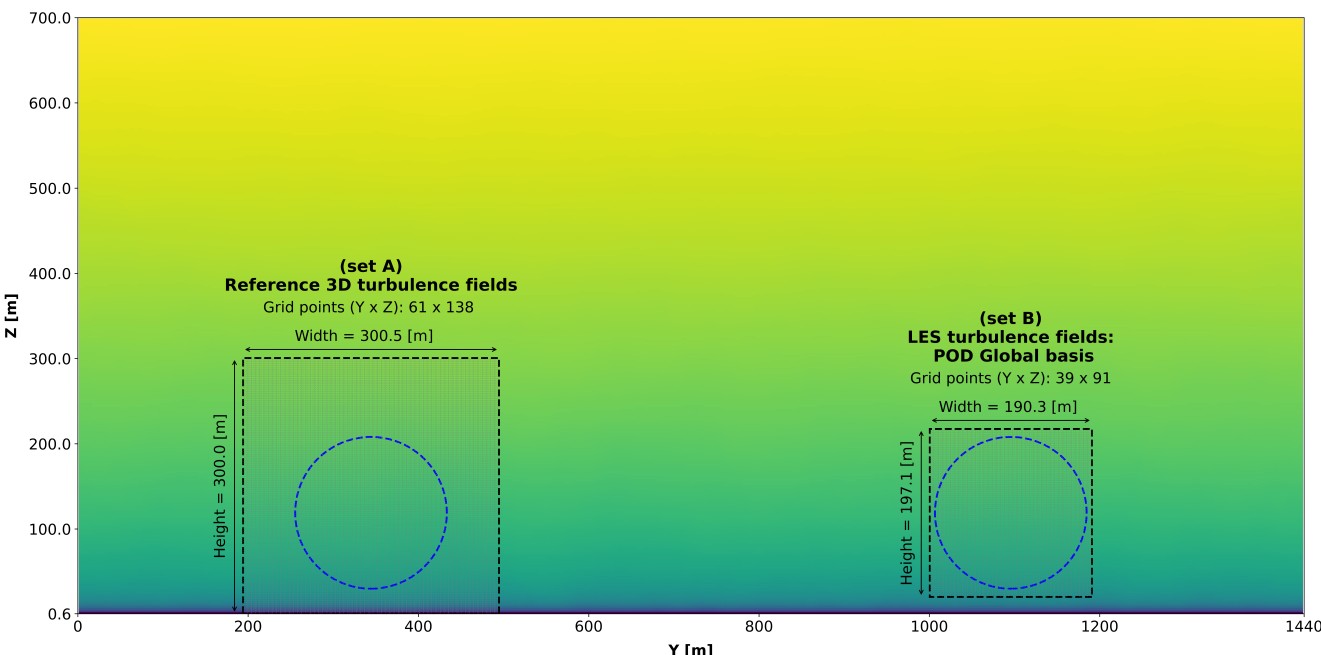

**Figure A1.** LES wind speed profile representation, and the two datasets used on this study, where set A represent the dataset from where the
3D turbulence fields were extracted, while set B represent the location where the dataset for the global POD modes generation was extracted.

## Appendix B:  Global POD modes

The global POD modes are derived from the inflow database detailed in Sect. 2.3. The first ten POD modes for the global basis
are shown in Fig. B1. These modes are ranked by decreasing total kinetic energy (TKE), revealing large-scale structures in
the lower-order modes that progressively diminish in size with increasing mode number. Overall, the resulting spatial modes
are consistent with those previously reported for both single-wake scenarios (Sørensen et al., 2015; Bastine et al., 2018) and
multiple-wake configurations (Andersen et al., 2013; Andersen and Murcia Leon, 2022).

This consistency highlights the similarity of dominant coherent structures across cases, demonstrating the potential for a
reduced-order model built upon these generic patterns (Céspedes Moreno et al., 2025). However, the importance (order) of



different modes will differ across different cases, where the low-frequency fluctuations are predominant, and they disappear at higher modes (Andersen and Murcia Leon, 2022).

It is important to note that these POD modes include only the longitudinal ($u$) fluctuating velocity component, since this is the reconstruction target in our study.

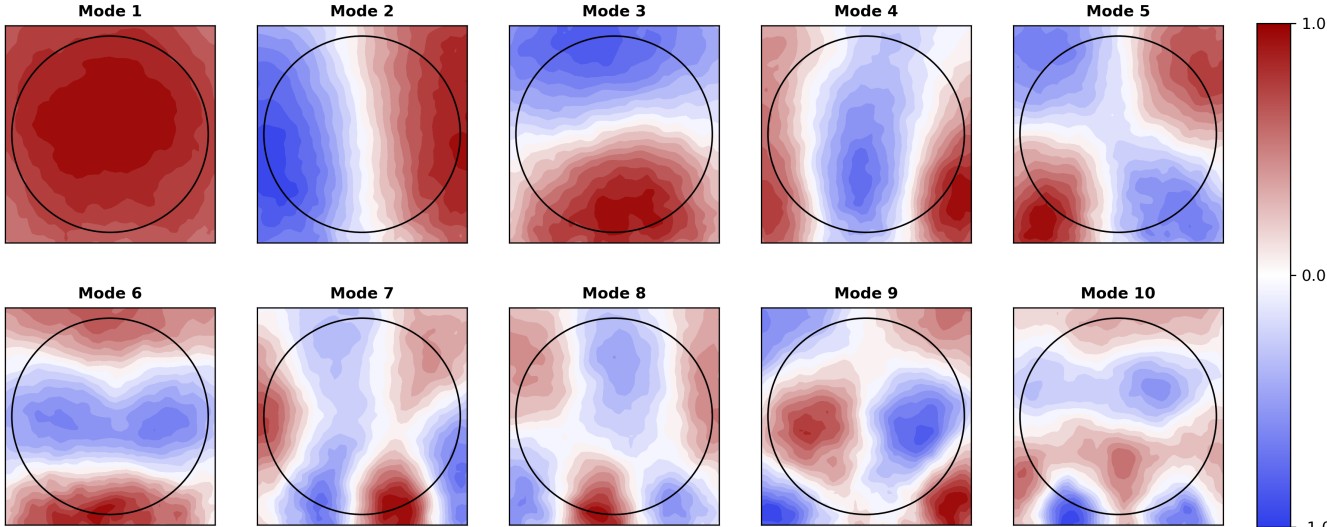

**Figure B1.** First 10 global POD modes, only estimated for u component.

## Appendix C: Time series

To illustrate the effect of increasing the number of scans on the different reconstruction methods, Fig. C1 presents time series at a single location in the $YZ$ plane for all evaluated methods, located at $y = -65.1\ m$ and $z = 156.1\ m$ (approximate 75% radius span). Two input cases are considered: the true wind speed ($u_{\mathrm{fw}}$, solid lines) and the volume-averaged lidar estimate ($u_{\mathrm{lidar,\ wgh}}$, dashed lines), both using a high scan count of $n_{\mathrm{scan}} = 16$. This high scan count is chosen to highlight the impact of longitudinal spatial filtering, not as a practical recommendation. Above each subplot, the corresponding MAE is reported, representing the time-averaged deviation of the reconstructed signal from the LES reference signal over the full 10-minute period.

The baseline method exhibits minimal differences between the two wind speed inputs, effectively capturing the average behavior of the time series but failing to reproduce high-frequency fluctuations. This limitation reflects the method's lack of spatial and temporal adaptability. In contrast, the POD-LSQ method closely follows the LES reference in both amplitude and phase when using the true wind speed ($u_{\mathrm{fw}}$). However, when using the lidar-based input ($u_{\mathrm{lidar,\ wgh}}$), it captures the high-frequency variations less accurately. This is caused by the high scan number, which results in a large longitudinal span of the lidar measurements used to fit the POD amplitudes, introducing a low-pass filtering effect in the time domain. In addition, the inherent probe volume averaging of the lidar further attenuates high-frequency fluctuations (Peña et al., 2017).

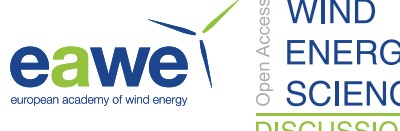

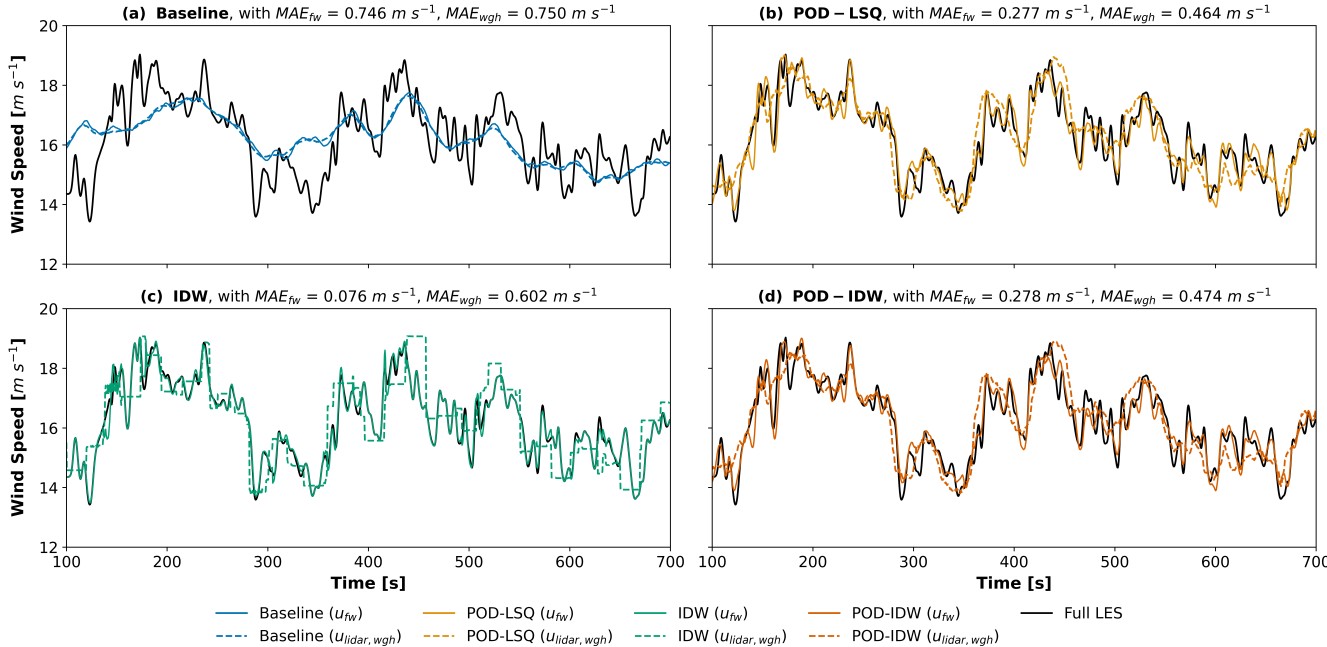

**Figure C1.** Comparison of 10-minute time series for each method, at a single location in $y = -65.1\ m, z = 156.1\ m$, using $n_{scan} = 16$, $\theta = 22.5°$, $K = 100$ global POD modes for POD-based methods as input parameters for the reconstruction, for the true wind speed ($u_{fw}$) and volume-averaged lidar estimate ($u_{lidar,wgh}$) cases.

The IDW method shows excellent agreement with the LES signal when using $u_{\text{fw}}$, nearly replicating the full temporal dynamics. Yet, with lidar-based input, its performance declines significantly, yielding blocky and discontinuous reconstructions that highlight its sensitivity to multi-distance fixed-plane mapping error. This is because the IDW method with $u_{\text{fw}}$ samples the true wind at the desired $YZ$ positions in the $X_{target}$ plane, whereas the values of $u_{\text{lidar, wgh}}$ used for interpolation are located at different longitudinal positions. Furthermore, the values for $u_{\text{fw}}$ used during the IDW interpolation are updated at every 655   time step of the simulation, whereas the values for $u_{\text{lidar, wgh}}$ do not change. Thus, the time series for IDW with $u_{\text{lidar, wgh}}$ has a "quantized" look, because the point in space is being interpolated from the same measurement points, until enough time has elapsed that a new, closer measurement has acquired.

    While POD-IDW performs slightly worse than POD-LSQ in terms of absolute error, it shows significantly improved robustness compared to IDW when using lidar-based inputs. By applying POD fitting on top of the interpolated IDW field, the 660   method mitigates the blocky and discontinuous behavior introduced by direct IDW interpolation—particularly the "quantized" appearance caused by fixed measurement locations over time. This improvement stems from the projection of the IDW field onto a set of spatial patterns derived from POD, which enforces spatial coherence and smooths out interpolation artifacts. As a result, POD-IDW produces more continuous and physically consistent time series, as further illustrated in Fig. C1d.



*Code and data availability.*

The hub-lidar database generated from LES inflow simulations is available at https://doi.org/10.11583/DTU.28151724
(Soto Sagredo et al., 2025a).

*Author contributions.*

ES, JR, SA, and AH contributed to the conception and design of the study. ES, with input and guidance from JR, developed
the HuLiDB framework, implemented and evaluated the wind field reconstruction algorithms, generated the numerical lidar
database, performed the analysis, and wrote the draft manuscript. SA generated and scaled the LES inflow data and extracted
the global POD modes. JR, SA, and AH provided support with the overall analysis and critically revised the manuscript.

*Competing interests.*

The authors declare that they have no conflict of interest.

**Acknowledgments**

This work is part of the CONTINUE project, which has received funding from the Danish Energy Technology Development
and Demonstration Programme (EUDP) under grant agreement no. 64022-496980. The authors gratefully acknowledge the
computational and data resources provided by the Technical University of Denmark through the Sophia HPC Cluster (2025)
(Technical University of Denmark, 2019). We extend our sincere thanks to Michael Courtney for his critical feedback and
insightful suggestions, which helped enhance the quality of this study. Finally, we acknowledge the use of OpenAI's ChatGPT
(GPT-4) to support improvements in grammar, clarity, and readability during the manuscript preparation (OpenAI, 2023).





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
