# Peer review of "Wind-field estimation for lidar-assisted control: A comparison of proper orthogonal decomposition and interpolation techniques"

_Wind Energy Science, 2025_

## Referee Comment (RC1)

**Peer review for WES-2025-148**

**Title**: "Wind-field estimation for lidar-assisted control: A comparison of proper orthogonal decomposition and interpolation techniques"

**Authors**:

- Esperanza Soto Sagredo
- Søren Juhl Andersen
- Ásta Hannesdóttir
- Jennifer Marie Rinker

**Overview:**

This manuscript presents a methodologically rigorous comparative study that addresses a well-defined research question in lidar assisted control for wind energy. The authors establish a clear central focus on evaluating three distinct wind field reconstruction methods—POD-LSQ, IDW, and POD-IDW—for real-time inflow characterization. The research demonstrates exceptional methodological consistency by testing all three approaches under standardized semi-realistic conditions using LES data, implementing a unified testing framework with HAWC2 simulations on the DTU 10MW reference turbine, and employing consistent performance metrics across all methods. The deliberate execution is further evidenced through comprehensive parametric analysis, systematically varying key parameters including scan count, POD mode number, lidar beam angle, and measurement uncertainty effects to fully characterize method performance under different operational conditions.

The authors appropriately frame the scope and applicability of the work by acknowledging testing under "idealized conditions" while still demonstrating practical relevance through semi-realistic LES-derived scenarios. The well-quantified results and balanced discussion of limitations suggest this study makes a solid contribution to lidar-assisted control methodology with clear practical applications in wind turbine technology. Given the breadth of cases tested in the study, it would be instructive to show a greater range of the MAE for the tested configurations. For example, response surfaces or heat maps that show MAE as a function of the number of scans and the lidar scan half angle would give the readers a clear picture of how much better the optimal point is than the suboptimal configurations, how convex the response is, etc. It would also be helpful to understand better the relationship between the PSDs of inflow wind with the final error metric used in the study.

Overall, this is an excellent contribution to the wind energy literature, building on a measured and consistent methodology to show the capacity of several inflow reconstruction methods to build on virtual lidar measurements. The results are framed in terms of downstream applications with some discussion of the limitations of the current work, and augmentations that would be needed for integration into functional lidar-assisted control. I recommend the authors review the comments and considerations below, intended to help readers get the most out of the work and position this research for future success.

**Comments:**

> Comment

> "The methods are tested under semi-realistic conditions derived from large-eddy simulations (LES), using a hub-mounted lidar sensor implemented in HAWC2 on the DTU 10 MW reference turbine."

> Comment

> "To address these limitations, Soto Sagredo et al. (2024a) proposed a least-squares fit of POD (POD-LSQ) method using hub-lidar data to estimate modal amplitudes in real time without requiring prior flow information."

> Comment

> "The primary goal is to quantify reconstruction error and analyze sensitivity to key input parameters: the lidar beam half-cone angle (θ), the number of scans (nscan), the number of POD modes used for truncation (K), and the influence of measurement uncertainty"

> Comment

> "The LES data used in this study originates from precursor simulations by Andersen and Murcia Leon (2022)."

> Comment

> "steady pressure gradient"

- Simulations are not driven through Coriolis forcing. I wonder how this impacts the development of sheer and veer in the simulated ABL.

**Comment**

> "The simulation assumes a surface roughness of zorg 0 = 0.05 m and a friction velocity of uo∗"

- What happens when the atmospheric conditions deviate from these boundary conditions? Is there any way to infer impacts to the inflow estimation quality?

**Comment**

> "neutral atmospheric boundary flows can be rescaled to generate multiple inflow conditions"

- Does this rescaling apply to the turbulent content of the inflow or is the mapping identical for mean velocity profile and turbulence?

**Comment**

> "Taylor's frozen turbulence hypothesis"

- In assessing the validity of Taylor's hypothesis in this case, what is the turbulence intensity for the flow? More importantly, what impact will the mean shear gradient and veer have on the resulting 3D turbulent inflow fields?
- Is a new global basis defined for each inflow scaled as per eq. 1, or is the global basis universal for all inflows with $u_{new} \in [8.0, 12.0, 15.0, 18.0] \text{ m s}^{-1}$

**Comment**

> "derived by combining multiple cases, providing a more general representation across a broader parameter space. As the global POD modes are derived from multiple flow conditions, they are not optimized for any single case but instead capture generalized flow structures across the parameter space."

- An important consideration when using a global basis of POD modes must be, "how representative is this basis for the flow in question?" Certainly we expect that modes defined from the ensemble of cases should be a good representation of the dynamics for each individual case unless there's some reason to question the statistical stationarity of the larger simulation. More generally, can the methods used in this work generalize to using the same global basis for cases where the background atmospheric changes have changed (e.g., for non-neutral conditions)?

**Comment**

> "A visualization of the first ten global POD modes is provided in Appendix B."

- The basis of global modes shown in Appendix B show behavior similar to 2D Fourier modes, consistent with a nearly uniform background flow. Is this expected? The text does not indicate the shear exponent or veer across the considered domain. These non-homogeneities would likely result in more complex behavior in the global modes.

**Comment**

> "Nevertheless, no motion 185 correction has been applied in the present analysis."

- Does the rotational velocity of the rotor also need to be accounted for in the motion correction or does considering only the $u$ component of velocity assume away the problem? I understand that no motion correction was applied in the end, but for completeness, it would be good to include a statement as to why it should be considered or can safely be neglected.

**Comment**

> "M"

- Should be $\mathbf{M}_{\text{lidar}}$?

**Comment**

> "Interpolation with IDW"

- This method shares some elements with that of Beck H, Kühn M. Temporal Up-Sampling of Planar Long-Range Doppler LiDAR Wind Speed Measurements Using Space-Time Conversion. Remote Sensing. 2019; 11(7):867. https://doi.org/10.3390/rs11070867.

**Comment**

> "except the baseline"

- It's not clear to me why the baseline method should miss the low-frequency content of the velocity spectra regardless of the number of scans considered, when all the other methods make a nearly perfect match. Additional interpretation would be appreciated.

Comment

> "Figure 7."

- Is there any way to relate the spectra to the MAE for each case? The spectra themselves describe the distribution of energy across a range of frequencies, but it's not clear how the missing energy in the mid-range or the extra energy for high frequencies might influence the MAE.

Comment

> "Tower natural frequency"

- Does the presence of the tower natural frequency in the inflow spectra suggest that the observations should be motion compensated? The peak is only visible in the virtual lidar measurements, not in the true wind speed.

Comment

> "Figure 8."

- Similar to the comment for FIgure 7, it's not clear how to consider the spectra in terms of the MAE. Results below suggest that the POD-IDW method produces lower values of MAE, even though the spectra in Figure 8 show a worse match to the Full LES spectra in the middle range of frequencies.

Comment

> ""...having a high number of spatially distributed inputs.""

- I think this is the real takeaway from Section 3.2 and 3.4—uniform information density across the domain is key. The more information you have to estimate the flow field (good coverage over the reconstruction area, minimal gaps or distance between useful bits of information) the better. This itself is not surprising, and there should be a good way of describing or estimating analytically the tradeoffs between the number of scans used, the value of interpolation, and the lidar beam half angle.

Comment

> "Future work"

- As a final suggestion, for future work it would be great to use this framework to describe the lidar measurement or scanning strategies to minimze error on the estimated inflow, or even better, in terms of granting control authority to the LAC system.